

# rSHUD v2.0: Advancing Unstructured Hydrological Modeling in the R Environment

Lele Shu[1, 2, 3], Paul Ullrich[2], Xianhong Meng[1,2], Christopher Duffy[4], Hao Chen[1,2,5], and Zhaoguo Li[1,2]

[1]Key Laboratory of Land Surface Process and Climate Change in Cold and Arid Regions, Northwest Institute of Eco-Environment and Resources, Chinese Academy of Sciences, Lanzhou, Gansu 730000, China

[2] University of Chinese Academy of Sciences, Beijing 101408, China

[3]Department of Land, Air, and Water Resources, University of California, Davis, Davis, California 95616, USA

[4]Department of Civil and Environmental Engineering, Pennsylvania State University, University Park, Pennsylvania 16802, USA

[5] College of Atmospheric Sciences, Lanzhou University, Lanzhou, Gansu 730000, China

**Correspondence:** Xianhong Meng (mxh@lzb.ac.cn) and Lele Shu (shulele@lzb.ac.cn)

## Contents





**Abstract.** Hydrological modeling is a crucial component in hydrology research, particularly for projecting future scenarios. However, achieving reproducibility and automation in distributed hydrological modeling research for modeling, simulation, and analysis is challenging. This paper introduces rSHUD v2.0, an innovative, open-source toolkit developed in the R environment to enhance the deployment and analysis of the Simulator for Hydrologic Unstructured Domains (SHUD). The SHUD is an integrated surface-subsurface hydrological model that employs a finite volume method to simulate hydrological processes at various scales. The rSHUD toolkit includes pre- and post-processing tools, facilitating reproducibility and automation in hydrological modeling. The utility of rSHUD is demonstrated through case studies of the Shale Hills Critical Zone Observatory in the USA and the Waerma Watershed in China. The rSHUD toolkit's ability to quickly and automatically deploy models while ensuring reproducibility has facilitated the implementation of the Global Hydrological Data Cloud (https://shuddata.com), a platform for automatic data processing and model deployment. This work represents a significant advancement in hydrological modeling, with implications for future scenario projections and spatial analysis.



# 1 Introduction

Scientific modeling utilizes mathematical methods of natural laws to predict unknown variables in space and time by incorporating known variables (Beven, 2012; Duffy, 2017). Hydrological modeling remains a crucial approach in hydrology research for analyzing future scenarios particularly. As hydrological models evolve from lumped to distributed models, the need for spatial data in modeling continually rises. Simultaneously, with the advancement of numerical methods as computation strategies in hydrological modeling, the models pose new challenges for hydrological unit partitioning (Paniconi and Putti, 2015; Peckham et al., 2017; Peel and McMahon, 2020).

Many hydrological models rely on GUI interface tools for data pre- and post-processing, such as arcSWAT (Arnold et al., 1998), PIHMgis (Kumar et al., 2010; Bhatt et al., 2014) and etc. Definitely, the GUI interface tools are user-friendly and easy to use, making them favorable for promoting models. However, there are two common challenges with GUI interface tools. First, poor repeatability due to difficulties in duplicating a user's data editing process leads to modeling discrepancies in the same simulation area. Second, GUI interface tools face difficulties in handling large amounts of modeling. In watershed modeling, human participation is required instead of being controlled via modeling parameters, making it impossible to implement large or automated modeling with GUI-based modeling tools. For instance, the deep learning methods' modeling process, which is mainly completed automatically via parameter configuration, allows 671 basins in CAMELS data to be automatically modeled, optimized, and verified (Newman et al., 2017; Beven, 2020; Nearing et al., 2021; Li et al., 2022). However, similar automated modeling, simulation, and analysis work with a distributed hydrological model seems impossible mission; considering that lumped models such as SAC-SMA (Burnash et al., 1973), which are frequently compared with deep learning models in the research literature, require minimal data pre-processing.

Therefore, both pre-processing and simulation post-processing require intervention-free, reproducible, and automated tools. Hence, the development of automated and reproducible pre-processing and post-processing tools for distributed hydrological models and numerical method hydrological models is imperative.

This paper introduces the rSHUD tool, an open-source toolkit developed in the R environment and supports pre- and post-processing functionalities for the Simulator of Hydrological Unstructured Domains (SHUD) and similar surface-subsurface integrated hydrological models. Its tools can be combined into automated processing scripts suitable for batch modeling, simulation, and analysis tasks.

The article begins by presenting the R environment and the rSHUD package in section 2, followed by a brief introduction of SHUD model structure in section 3. Then the critical steps in deploying the SHUD model using rSHUD tools are described in section 4. Lastly, in section 5, the paper showcases two basins as case studies to demonstrate rSHUD's modeling and results analysis process.

# 2 R and rSHUD

The programming language R is extensively used for analyzing data and conducting statistical computations. R is freely available and widely used by statisticians and scientists. R is a dynamically typed interpreted language that is primarily used





interactively due to the wide range of built-in core functions and libraries available. Users can define their own functions and procedures in R, C/C++, Python or Fortran due to the language's extensibility. R's efficient libraries and user-friendly design make it a leading resource in scientific computing.

R is cross-platform compatible, supporting Linux/Unix, Mac OS X, and Windows operating systems. R applications require the installation of a user interface, and one of the most commonly used R-specific graphical interfaces is RStudio (https://rstudio.com, accessed June 2023).

In addition to R's core libraries, users can add additional libraries for specific purposes, which is relatively simple and convenient. Standard functions such as *install.packages* can download and install a specific library from the CRAN repository. In addition, users can install developer libraries hosted on GitHub using the *devtools:install_github* function in the *devtools* extension library. Functions, *library* and *require* can load any installed packages into the working environment.

rSHUD is an open-source project available on GitHub (https://github.com/shud-system/rSHUD, accessed June 2023) and regularly updated. Since it is not yet available in CRAN's repository, users can install rSHUD and the necessary libraries in their environment using the *devtools* library. These two commands install rSHUD and the required libraries in the user environment. The following will install the required libraries and rSHUD in the user environment.

The rSHUD version matches the SHUD model version. The latest SHUD is version 2.0 (Shu, 2023a), so rSHUD is version 2.0 different from previous version 1.0 (Shu, 2020a).

```
install.packages("devtools")
devtools::install_github("SHUD-System/rSHUD")
```

Installation of rSHUD and the dependent library may take some time. The additional R libraries required for rSHUD are listed in Table A1 of appendix. The rSHUD package serves several important purposes in the field of geospatial data analysis and hydrological modeling. These include:

– Convert geospatial data to SHUD format. This package is equipped with a toolkit that handles raster and vector data. It then constructs an unstructured triangular mesh domain, which is an important step in preparing SHUD model data.

– Parameterization of the hydraulic properties of soil and land cover. This feature enables users to define and adjust the hydraulic properties of various soil types and land cover classes, improving the applicability of the model.

– The ability to read and write SHUD input files. This feature ensures smooth integration with the SHUD model and allows users to import and export data easily.

– The ability to load the output results generated by the SHUD model. This facilitates the interpretation and further analysis of the model's results.

– Facilitate hydrological time series and geospatial analysis. This feature allows users to perform detailed temporal and spatial analyzes of hydrological data, providing valuable insight into water dynamics.





– Exploring time series and spatial data. This function allows users to generate clear and informative visual representations of their data and helps with data interpretation and communication.

Each function in rSHUD has its own help page that provides information on its usage, arguments and return values. With 167 functions included in rSHUD, it is not feasible to provide explanations of all functions in this paper. However, users can access the help page for each function by using the command *help(thefunction)*.

## 3    Model and Data

### 3.1    SHUD

SHUD is a distributed hydrological model that is based on physical principles, namely, the Saint Venant Equation for the surface runoff and Darcy-Richards Equation for the subsurface flow (Shu et al., 2020). It solves the partial differential equations (PDEs) of hydrology with the finite volume method (FVM), which allows for the direct coupling of equations representing groundwater, soil moisture, surface water, vegetation, and land cover interactions. The model's nomenclature, Simulator of Hydrological Unstructured Domains, stems from this intricate process of domain decomposition and the subsequent numerical solution of the associated PDEs within these unstructured hydrological domains. SHUD is an improvement and revision of the Penn State Integrated Hydrologic Model (PIHM) (Qu and Duffy, 2007; Kumar et al., 2010; Bhatt et al., 2014; Yu et al., 2014). Shu et al. (2020) provides a detailed explanation of the differences between the two models.

SHUD offers flexibility in terms of time and spatial resolution. The spatial resolution of the model ranges from centimeters to kilometers, depending on the modeling needs and computational resources available. The internal time step, specified by the user as the maximum time step, can be adjusted, while the computing time step is limited to a few seconds. The model exports the status of the catchment at regular time intervals, ranging from minutes to days. Due to its flexibility, the model can be coupled with other systems such as agriculture, cryosphere, ecology and natural disasters. The SHUD model comprises two types of cells: hill-slope and river cells. In the planar view, the hill-slope cell has an unstructured triangular shape, while the river cell is portrayed as a rectangle. In the 3D view, the hill-slope cell is a triangular prism, and the river cell is a trapezoidal prism (Figure 1). Both the hill-slope and river cells are hydrological computing units (HCUs) in the SHUD numerical sover.

### 3.1.1    Hill-slope cell

The SHUD model utilizes unstructured grids to represent the computing domain of slopes. By default, the Denauley triangle method is used for building the domain, although other triangular network generation methods are also acceptable. Each triangle comprises three nodes with unique three-dimensional coordinates ($x$, $y$, $z$ in meters) that define their location. The centroids of the cells are calculated within the SHUD program. Topological relations are critical to the model and describe the one to three neighbors and nodes associated with each triangle.





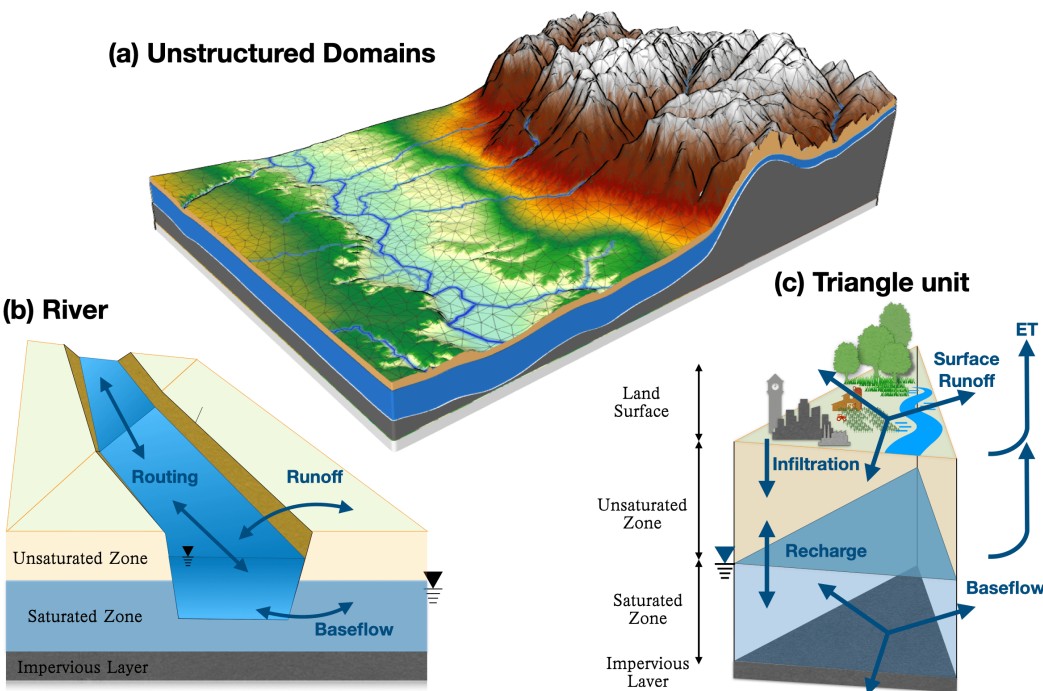

**Figure 1.** Spatial and hydrological structure of the SHUD model. Unstructured domains in the partial watershed (a), River cells and topological relationships with triangular cells (b), and 3-layer triangular prisms(c).

### 3.1.2 River cell

In geospace, a river network is a series of sequentially connected polylines that are defined as an ordered set of nodes in three-dimensional coordinates. A river reach is a polyline between two critical nodes. A critical node can be the intersection of multiple rivers or a user-assigned point. Ordinarily, the first node marks the beginning of a reach, while the last node represents its end. The Strahler stream order (Strahler, 1952) in SHUD determines the order of branching of the river system. The triangular domain in SHUD overlays the river network, and the topological relationship between the polylines and triangles

determines the exchange of surface and subsurface water between the river channel and hill-slope (Shu et al., 2020). The river outlets are typically located at the edge of the watershed.

The topological relationships between the river and hill-slope cells are an essential difference between SHUD and PIHM. In the PIHM model, the river network is adjacent to two triangular cells, which results in three issues:

1. The length of the river network has an important impact on the number of computing units in both river and hill-slope
domain. Users need to balance river channel simplicity with the number of computation units; the simplicity implies modification on natural feature of river network.





2. In plain areas, the heavily meandering river network generates hundreds of small triangular cells, easily exceeding the necessary number of HCUs and slowing down model computation dramatically and unnecessarily.

3. The accumulation of water in sink cells violates the fundamental assumption of the shallow surface runoff equation, consequentially, cause the model to become unstable and result in poor performance. At the start point of first-order stream, the occurrence of local sink points are very often and greatly reduce the computational performance of the entire watershed.

To tackle the above issues in PIHM, a modeler need to manually modify the shape of the river channels repeatedly, which reduces the efficiency and reproducibility of modeling. Therefore, SHUD's river network overlays on triangular cells and the exchange of water between hill-slopes and the river network is calculated based on the water level of the river channel and the slope of underground water and surface water (Shu et al., 2020).

### 3.1.3 Vertical layers

SHUD defines three vertical layers for each hill-slope cell (Figure 1(c)): the land surface, the unsaturated layer (vadose zone), and the saturated layer (groundwater). By default, the model assumes that the no-flow boundary at the bottom is an impervious layer, also known as the impervious bedrock layer.

Thus, the model's default settings define three elevations: the land surface ($z_s$), the groundwater table ($z_g$), and the bedrock ($z_b$). The water stored above $z_s$ is called surface ponding water, while the water between $z_g$ and $z_b$ is known as groundwater. The elevations of $z_s$ and $z_b$ remain constant for each cell, whereas $z_g$ fluctuates based on groundwater storage. The space between $z_s$ and $z_g$ is known as the vadose zone. If $z_g$ rises enough to meet $z_s$, then the vadose zone disappears.

## 3.2 Rawdata

Hydrological models are a combination function that accepts certain input data and parameters and then produces their output. For modelers, the first question is: what data do we need for a hydrological model and what results do we get from the model?

The rSHUD package utilizes three primary types of data: spatial data, time-series data, and attribute data.

– **Spatial Data:** This encompasses a variety of geospatial elements such as elevation, soil classes, land cover classes, meteorological stations/coverage, the boundary of a research area, and the stream network. Spatial data can be in either vector or raster format, both of which effectively represent spatial heterogeneity.

– **Time-Series Data (TSD):** This includes meteorological data (precipitation, temperature, radiation, wind speed, and humidity) and phenological data (leaf area index and snow melt factor). Hydrologic models, depending on their conceptual and mathematical scheme, are sensitive to the time interval of the forcing (or meteorological) data. For instance, the intensity, duration, and frequency (IDF) of precipitation are critical to models for flood prediction, soil erosion, pollutant monitoring, and so forth. Therefore, for short-term hydrologic events, hourly to daily meteorological data is preferred in process-based models, while long-term trend modeling accepts daily to yearly data in most conceptual or water-balance





models. Given that the SHUD model employs physical hydraulic equations and depicts hydrologic processes in fine spatial and temporal scales, it is recommended to use sub-daily meteorological data.

For calibration and validation purposes, the model also requires reference data, which could be observational data, comparable datasets from other models, or data from previous publications.

– **Attribute Data:** This includes the feature of the spatial data, used for generating hydraulic parameters of each soil class, geologic layers, and land cover classes. The required soil texture parameters include the percentage of silt and clay, organic matter content, and bulk density. These parameters are used to calculate the hydraulic parameters, including
hydraulic conductivity, porosity, and the $\alpha$ and $\beta$ values in the van Genuchten equations (Shu et al., 2020).

## 4   Model deployment

Deploying a hydrologic model involves several basic steps. Figure 2 illustrates the typical workflow of hydrologic modeling in a research region. First, data preparation is required to build a dataset subset for the research area. Second, data pre-processing is necessary to reformat the spatial data and attribute table. Third, the model must be built, generating input files for the hydrologic
model. Fourth, the program must be executed to perform the modeling. Finally, post-processing is to read the output files and analyze the results.

The rSHUD package aims to support the three out of five steps in the model deployment: data pre-processing, building model and post-processing.

### 4.1   Data pre-processing

Multiple data processing stages were involved in this step. We removed holes, projected the modeling boundaries, and generated buffer zones in a series of steps. Additionally, we removed irrelevant data from the Digital Elevation Model (DEM) data, extracting only the relevant data within the study area. We then reprojected and simplified the DEM data as a Projected Coordinate System for ease of analysis. We verified and corrected the river flow direction consistency for the river network data, removed duplicate points and segments, and standardized the data format.

The data pre-processing stage included reformatting the spatial data into a consistent format and generating hydraulic parameters based on the land cover, soil, and geology classification. Prior to data pre-processing, sufficient spatial data were ready, including DEM, soil, land cover, watershed boundary, and river network. As the SHUD model requires a specific format for the time-series data, the forcing and phenology data format and units must be standardized.

Consistency of spatial coordination in data processing is indispensable to ensure accurate and reliable results. It is necessary
for spatial data to have uniform projection information, which basically means a defined coordinate system. Analysis tools usually accept data either from a Geographic Coordinate System (GCS) or a Projected Coordinate System (PCS), a method to represent GCS on a flat surface. However, for hydrological modeling, data in different projections should be re-projected into a specific PCS before spatial processing. This is important since crucial spatial information from maps, including distance





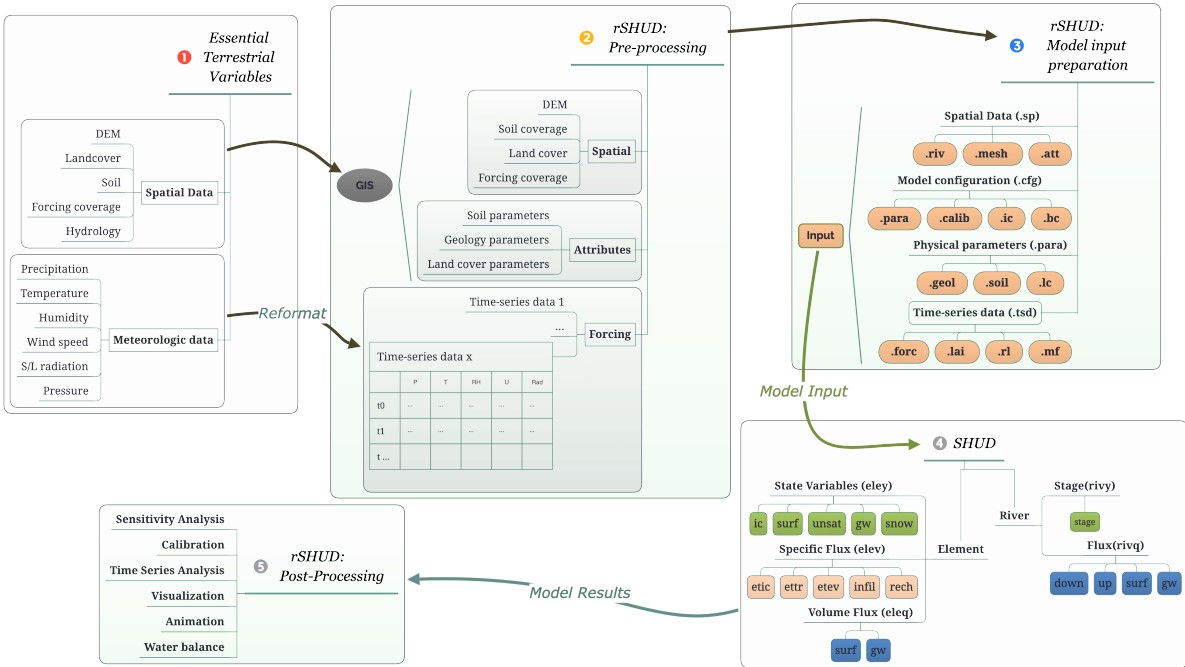

**Figure 2.** The general workflow of hydrological modeling and implementation of SHUD modeling system, which includes five steps: raw data accessing, pre-processing, building model

(in meters), direction, and area (in square meters), varies across different PCS. It is necessary to ensure that data from various

sources overlay correctly since spatial inconsistencies can occur.

It is recommended to use the Albers Equal Area projection, which has two reference latitudes and one central longitude. The reference latitudes are set at 1/4 and 3/4 of the watershed latitude range, respectively. Additionally, the central longitude corresponds to the longitude of the watershed centroid. An example is provided in section 5.

Typically, the extent of the raw spatial data exceeds the watershed boundary; therefore, it is necessary to subset the data.

Moreover, the final boundary of the modeling domain may differ from the original watershed boundary because spatial processing simplifies it. Thus, the subset should be slightly larger than the research area. A enclosure mask layer is generated from the watershed boundary with a buffer distance.

The original attributes in the soil data include soil texture components such as silt, sand, and clay percentages, as well as bulk density and organic matter content. To derive hydraulic parameters such as hydraulic conductivity, porosity, and van

Genuchten parameters, a pedotransfer function is used based on the soil texture data (Wösten et al., 2001; Shu et al., 2020). The pedotransfer function used in rSHUD is listed in Appendix B.





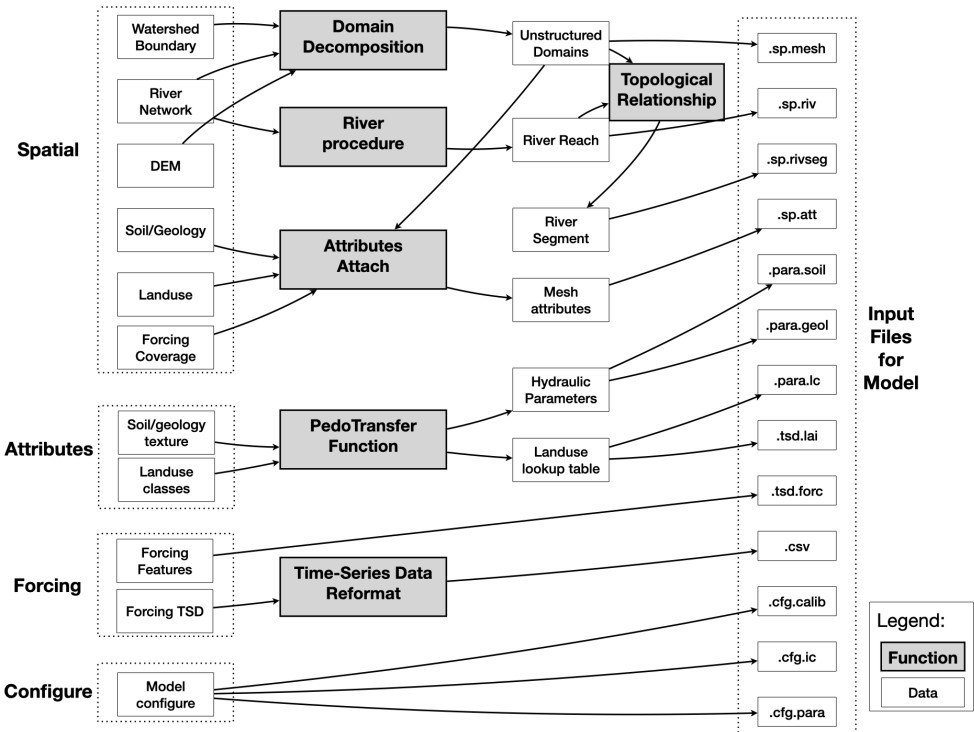

**Figure 3.** The procedures of model building and the outputs files of rSHUD. The left dash boxes are input rawdata and the right dash box are model input files.

## 4.2 Model build

### 4.2.1 Domain Decomposition

Domain decomposition is a mathematical strategy that employs geometric principles to partition a larger domain into smaller, more manageable subdomains. Within the context of the SHUD model, this approach is implemented to decompose the watershed into a Triangular Irregular Network (TIN). As a result, the watershed is composed of a collection of unstructured triangles.

The SHUD model can use regular triangles as the computational domains, but unstructured domains are recommended. Unstructured domains offer more flexibility to represent the irregular watershed boundary and terrain features. Moreover, unstructured domains allow for different resolutions within a watershed. For example, modelers can apply coarse TINs in the whole watershed but finer TINs in the river corridor when the research topic concerns surface-subsurface water exchange along the corridor. The multi-resolution configuration ensures sufficient resolution in the area of interest while maintaining the mass balance of the watershed, without increasing the number of HCUs excessively as in regular grid domains.





The triangulation method *shud.triangle* in rSHUD package is from *RTriangle*, an R port of Jonathan Shewchuk's Triangle
library (Shewchuk, 1996, 2002). The *RTriangle* generates Delaunay triangles without no small or large angles and is thus
suitable for finite element/volume analysis. The *shud.triangle* function requires watershed boundary as the mandatory input
argument while DEM, river network, islands/holes and suggesting points are optional conditions to constrain the triangulation.
The additional spatial data inserted during the triangulation process allows users to adjust the resolution of HCU in different
areas according to their research needs and study area characteristics, enhancing the flexibility of domain decomposition. There
are three extra arguments in *RTriangle::triangulate()* that are useful to build the ideal domains: minimum angle of a triangle
(argument *q*), maximum of triangle's area (argument *a*)and the ideal number of triangles(argument *S*).

Prior to performing triangulation, the boundary must be simplified using a tolerance. This process aids in smoothing the watershed boundary while ensuring an appropriate number of HCUs. For instance, when applying watershed delineation methods
to retrieve the boundary of a watershed using a 30-meter DEM, the resulting boundary often exhibits jagged 30-meter scale
edges. Therefore, in triangulation, the maximum edge of a triangle on the boundary is limited to 30 meters, while the inner portion of the mesh domain is composed of larger triangles. Consequently, The substantial difference in triangle area between the
edges and inner parts significantly slows down model performance by unnecessarily increasing computations on the boundary.
This negative impact on model efficiency must be addressed by simplifying the boundaries. Simplification is also necessary for
additional triangulation constraints such as a river, lake, and urban areas.

After triangulation, the *shud.mesh* function generates a mesh file that defines triangles by the index of three vertices and
three neighbors and the definition of all vertices in the domains ($x$, $y$, $z$ coordinates). The definition of TINs can be converted
into a text mesh file by *shud.mesh*, which inversely can then be interpreted into a Shapefile of polygons by a GIS function
called *sp.mesh2Shape*. These functions can cross-validate the consistency of the mesh domain.

### 4.2.2 River

The processing for the river is to 1) simplify the river reaches, 2) build the flow path, 3) determine the river order, 4) extract the
slope characteristics and 5) describe the geometry of river reach. These functions are integrated into the *shud.river* function,
the functions also can be called independently.

1. The simplification of the river reaches includes two meanings: the one is to simplify the meandering rivers straight,
   which is optional in SHUD; the other is to cut the very-long river reaches into smaller pieces.

2. In GIS and geomorphology, building a river path involves building the connections between upstream and downstream
   river lines. These connections form the river network and represent how water flows along the river lines. The direction of
   the river lines determines the inflow and outflow between two rivers. The function is realized by combining *sp.RiverDown*
   and *sp.RiverPath*.

3. River order is a categorization method used to understand the structure of a river network based on the levels of branching
   within it. The Strahler method (Strahler, 1952) is a wide approach for calculating river order. It assigns a value of 1 to all
   headwater streams and then increments the value by one for each confluence of two streams of equal order. In cases where





two streams of different order join, the order of the downstream stream is used. This process is repeated downstream until all streams have an order assigned to them. The calculation of river order based on the Strahler method is performed by using the function *sp.RiverOrder*.

The order of river reaches determines the generalized geometry (definition of the trapezoidal shape) and hydraulic parameters (Manning's roughness, Chezy coefficient, and so on) of each river reach. In the default configuration, all river reaches in the same order share the same geometry and hydraulic parameters. When a detailed description of individual river reaches is available, the model can also accept a detailed description for each river reach in the model domain instead of categorization by river order.

4. The slope is a critical parameter used to calculate the fluxes in river routing. It is determined by calculating the gradient between the elevations and distances of the starting and ending points of a river reach.

5. The default geometry of a river cross-section is represented by a trapezoidal shape, defined by the width of the riverbed ($w$), the slope of the bank ($s_b$), and the depth of the river channel ($D$) (Figure 4). The trapezoidal shape is very flexible and can be simplified to a shape of a rectangle ($s_b = 0$) or a triangle ($w = 0$), depending on the specific case. The function

*RiverType* generates a default table of river geometry and hydraulic parameters, which modelers can modify based on their own data.

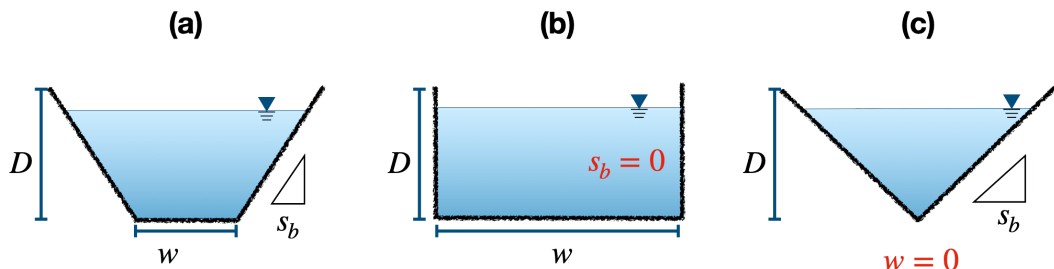

**Figure 4.** The cross-section of river in the SHUD model, which is described with the width of the riverbed ($w$), the slope of the bank ($s_b$), and the depth of the river channel ($D$). The trapezoidal shape (a) can be simplified to a shape of a rectangle (b) or a triangle (c).

The SHUD model determines the surface/subsurface water fluxes connection between the river and hill-slope cells based on their overlapping relationship. However, the topological relationship between triangles and rivers does not match perfectly. Specifically, a river reach overlays with multiple triangles and exchanges water with these triangular cells. Therefore, to account

for the exchange of water, each river reach is divided into several segments, each exchanging water with one triangular cell. The properties of each river segment include the index of the belonging reach, the index of the overlapping triangular cell, and the length of the segment within the cell. The function *sp.RiverSeg* is used to carry out this task.




### 4.2.3 Cell Attributes

The attribute file contains information about the features of each triangular cell, which is used for hydrological parameters. An
example can be found in Figure 5. The index of soil is used in the *.att* file to specify a row of parameters in the *.para.soil* file.
Similar to the soil index, the index for geology, land cover and forcing in the attribute file points to a row in the *.para.geol*,
*.para.lc*, and *.tsd.forc* files respectively.

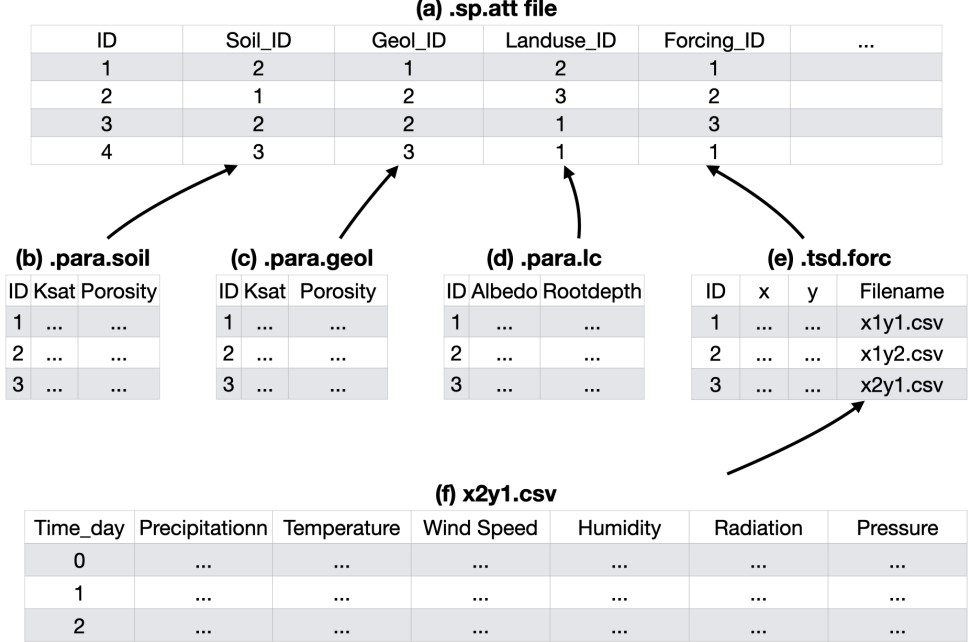

**Figure 5.** The structure of the model input files and their logical connections. Files are (a) attributes file (.sp.att), (b) hydraulic parameter of
soil (.para.soil), (c) geology (.para.geol), (d) land cover (.para.lc), (e) time-series files of all forcing data (.tsd.forc), and (f) the time-series
data for specific sites (.csv)

The *shud.att* function extracts the spatial data index as the properties of each cell. It extracts the features of multiple layers
through the centroids of the triangular cells, thus capturing only the value of the centroid instead of the mean or statistical value
of the entire cell. For example, an cell may contain only one soil type with no heterogeneity within the cell. The diversity of
soil properties among cells expresses the hydrological heterogeneity in a watershed. Hence, to represent high heterogeneity, a
finer resolution of the domain decomposition is required.





### 4.2.4 Hydraulic parameters

Three main files provide the necessary hydraulic parameters for hydrological simulation: soil, geology, and land cover. The
pedotransfer function (Wösten et al., 2001) utilizes soil texture to derive the hydraulic parameters, including vertical and
horizontal hydraulic conductivity and porosity, for the deep groundwater layer. The Appendix B lists the equations of the
pedotransfer functions developed by Wösten et al. (2001).

1. The term "soil" specifically refers to the top layer of soil that influences the vertical flux of water from the surface to the
   soil and the deep recharge to the saturated layer. The soil layer's essential physical parameters include vertical saturated
conductivity, porosity, residual water content, and values $\alpha$ and $\beta$ for the van Genuchten equation (Shu et al., 2020).

2. The SHUD model's geology layer refers to the aquifer profile's deeper layer. This layer's parameters describe the prop-
   erties of saturated groundwater flow, and the crucial parameters include vertical and horizontal hydraulic conductivity
   and porosity.

3. The hydraulic parameters of land cover, such as vegetation root depth, soil degradation ratio, imperviousness factor,
and Manning roughness, are stored in a lookup table within the data repository and can be easily transferred to a user's
   dataset for their modeling efforts. However, it is necessary to note that the lookup table is specifically designed for a
   particular land cover classification and is not transferable between different classification systems. For instance, the table
   for National Land Cover Datasets (Wickham et al., 2020) is not applicable to USGS 0.5-km MODIS-based Global Land
   Cover Dataset (Broxton et al., 2014) or China Land Cover Dataset (Yang and Huang, 2021). Appendix A provides more
insight into the default values for multiple land cover classification systems. The University of Maryland (UMD) Global
   Land Cover Classification (Hansen et al., 2000) determines the parameters for typical land covers based on which the
   values for other classification systems can be adjusted and transferred. The parameters other than UMD are transferred
   from UMD data(Hansen et al., 2000; Bhatt, 2012). However, the values in the tables are somewhat arbitrary and act as
   preliminary values that need to be updated when users have more reliable data.

### 4.2.5 Time-Series Data

The *.tsd.forc* file (Figure 5) saves a table about the forcing sites, including the $x$, $y$, and $z$ of the sites and the time-series
filename of it (Figure 5(f)). The SHUD program read the time-series file during the model simulation.

The phenology in SHUD is LAI's TSD, indicating growth, prosperity, and withering. The TSD is also a experiential value
for each land cover class from UMD vegetation parameter values(last access on June 2023, https://ldas.gsfc.nasa.gov/nldas/
vegetation-parameters, accessed June 2023). Similar to the lookup table of land cover parameters, the LAI TSD can be replaced
by the user's local observation data.

As the melting factor in SHUD is experiential monthly values for the Degree-Day model calculating the snow melting flux
(Hock, 2003; Zhang et al., 2012), the function of *MeltingFactor* adapt the following equation from Bhatt (2012):

$$M_f = \left\{ \left( \frac{M_{max} + M_{min}}{2} \right) + \sin \left( \frac{2\pi N}{366} \right) \times \left( \frac{M_{max} - M_{min}}{2} \right) \right\} \times 0.4 \tag{1}$$





$M_f$ is the melting factor used in the Degree-Day model $[mm\ day^{-1}K^{-1}]$. The maximum and minimum values of the melting factor during a year are represented by $M_{max}$ and $M_{min}$, respectively, and occur on June 21st and December 21st. $N$ reflects the Julian day difference from September 21st. As a result, the melting factor follows a sine curve, with its maximum on June 21st and minimum on December 21st.

Besides, there are boundary conditions (*.tsd.bc*) and observation data (*.tsd.obs*) in TSD format, which is optional for the model simulation. The boundary condition may be the irrigation, pumping, leaking, or known water management practices in time series. The optional observation data (*.tsd.obs*) is generally used for the model calibration only.

### 4.2.6 Model configure

In addition to the three primary categories of data – spatial, time-series, and attribute data – that hydrologic models commonly require, specific models may also necessitate the inclusion of configuration files. These files define various aspects of the model's operation, establishing the model's running range, determining the level of computational precision, setting the computing/exporting time-step, and specifying the file format, among other parameters. Thus, these configuration files provide a means to customize the model's functionality and output to meet specific research tasks and objectives.

The SHUD configuration includes four files: configuration of simulation (*.cfg.para*), calibration file (*.cfg.calib*), the initial condition of simulation (*.cfg.ic*) and boundary condition index (*.cfg.bc*). To generate boundary conditions with *shud.ic()* function, it is necessary to know the index of triangular meshes and river reaches. The indexes in boundary condition point to specific indexes in the TSD BC files. The initial conditions for river reaches include the initial water level in the channel. Details and file structures of these files are described in the SHUD manual (https://shud.xyz/book_en , accessed June 2023) (Shu, 2019).

## 5 Examples

To demonstrate the workflow of using the rSHUD package, we chose two watersheds as case studies and gradually implemented the processes of data pre- and post-processing of results. We selected the Shale Hills Critical Zone Observatory (SHCZO) in the USA and the Waerma Watershed in China. The rSHUD package already includes all the raw data needed for building the hydrological model, so there is no need to download extra files.

The R scripts for these exemplary watersheds can be found in the appendices C1, C2, and C3. As the SHCZO is a small and simple catchment, we created a script for the deployment of the SHUD model briefly. The other two scripts (Appendix C2, and C3) are relatively sophisticated for pre- and post-processing the SHUD modeling in the Waerma Watershed. All these scripts are embedded in the rSHUD package already. The files *demo_sh.R* and *demo_waerma.R* in the rSHUD source code is used to deploy SHUD in the SHCZO and Waerma, respectively, and *demo_waerma_ana.R* is used for the post-processing of the Waerma Watershed. The R scripts are self-explanatory, and users can read the annotations and understand the functionality of the codes; Therefore, we have omitted the details in this paper.



## 5.1 Shale Hills Critical Zone Observatory in the USA

The Shale Hills Critical Zone Observatory (SHCZO) is a small ($\approx 84{,}000 \ m^2$), forested catchment located in central Pennsylvania, USA. Its topography boasts relatively steep slopes (>0.18) and narrow ridges leading to its Shaver Creek tributary. The elevation of the catchment varies from 250 to 320 meters, and it experiences a humid continental climate that averages at 9.5
$^\circ C$. SHCZO receives an annual mean relative humidity of 65.2% and precipitation of around 1028 $mm$. Evapotranspiration is estimated to be 59.4 $mm$ with an annual runoff of 509 $mm$, translating to a runoff ratio of about 49.6%.

All data for the SHCZO modeling is downloaded from the Critical Zone Observatory website ( https://czo-archive.criticalzone. org/shale-hills/data/, accessed June 2023). The DEM is 1-meter LIDAR data, and soil is from the local survey (Lin, 2006; Yu et al., 2014). The watershed boundary and river network is calculated from the watershed delineation algorithm in PIHMgis
(Bhatt et al., 2014). The local meteorological station provides the forcing variables.

Due to the availability of high-resolution data in this SHCZO and the watershed's small size, a high-resolution SHUD model will be constructed. Since the SHUD model implements a triangular mesh, the triangle's dimensions will vary. We anticipate a maximum triangle area of approximately 200 $m^2$. In the result, the function *shud.triangle* with the constraints produced 698 triangles in SHCZO (Figure 6). The area of cells within the mesh followed a normal distribution, with values ranging between
40 and 200 $m^2$. This distribution highlighted the presence of both small and larger triangles within the model.

The remaining R script of *demo_sh.R* requires no elaboration as it is easy to read. The code saves all data in a user-specified path. Once data preparation is complete, the SHUD model can be compiled and run. We named the SHCZO project as *sh*; to initiate the simulation on Linux-like platforms, the user should enter *./shud sh*. In this example, the forcing data covers two years (2008-2009), and the simulation took about 5 minutes to complete on an Intel 6-Core I5, 64G RAM computer.

## 385 5.2 Waerma in China

The Waerma watershed is a headwater of the Yellow River, with an area of 9.8 $km^2$, located near Waerma Village, about 20 kilometers northwest of Maqu County in Gansu province of China. It has an elevation of 3800-4500 $m$ with significant terrain fluctuations and an average slope of about 0.42 (rise versus distance). The annual average temperature is about 1.2 $^\circ C$, and the annual average rainfall is about 630 $mm$. The main vegetation types are grasslands, meadows, and shrubs. The Key Laboratory
of Land Surface Process and Climate Change in Cold and Arid Regions, Chinese Academy of Science, built a comprehensive and detailed observational system for meteorology, land process, hydrology and cryosphere in the Waerma watershed (Meng et al., 2023).

The expected modeling configuration is the following: using the CMFD data 2000-2001 to drive the SHUD model with larger than 150 $m$ spatial resolution. The rawdata is described in Table 1. These data also can be retrieved via the Global
Hydrologic Data Cloud (https://shuddata.com, accessed June 2023) (Shu, 2023b).

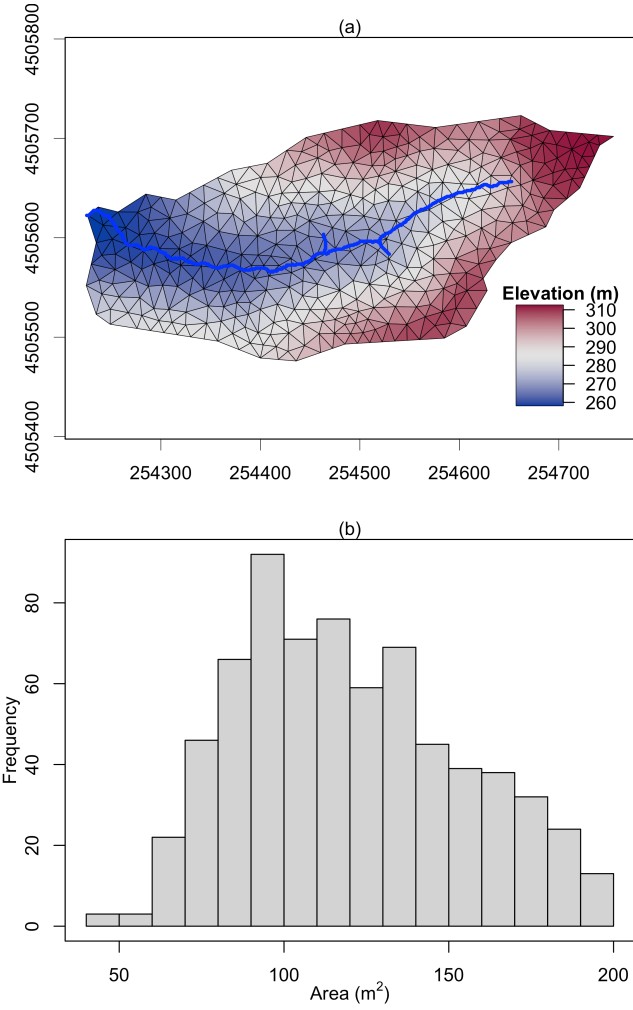

**Figure 6.** The triangular mesh (a) generated by the function *shud.triangle* for Shale Hills Critical Zone Observatory (SHCZO), and the histogram of triangles' area (b). The color in plot (a) is the centroid elevation of triangles.

### 5.2.1 Deployment

The script to deploy SHUD in Waerma Watershed is saved in *demo_waerma.R* and the watershed data is also available in the source code package. Following are the steps to deploy the SHUD model in Waerma Watershed by rSHUD.

1. Load necessary R libraries.

2. Setup and create folders for exporting data and figures.

3. Setup the environment for rSHUD.





| Data | Description | Source |
|------|-------------|--------|
| DEM | 30m | ASTER Global DEM (NASA/METI/AIST/Japan Spacesystems and U.S./Japan ASTER Science Team |
| Watershed Boundary | - | Generated from DEM delineation |
| River Network | - | Generated from DEM delineation |
| land cover | 0.5 km | USGS MODIS land cover data (Broxton et al., 2014) |
| Soil | 1 km | Harmonized World Soil Database v1.2 (Nachtergaele et al., 2008) |
| Forcing | 0.1 deg, 3-hr interval | China Meteorological Forcing Dataset (CMFD) (He et al., 2020) |

**Table 1.** The description and source of rawdata for Waerma Watershed.

4. Read/load this project's raw spatial and attribute data. All the spatial data must be reprojected to an identical PCS in this step.

5. Configure the modeling parameters, including the maximum triangles' area, the minimum angle of triangles, tolerance to simplify watershed boundary and river network, thickness of the aquifer, and number of days of simulation periods.

   The expected minimum resolution of modeling in Waerma is 150 $m$; therefore, in triangular mesh, the maximum cell area is about 225000 $m^2$. After the domain decomposition with *shud.shud.triangle*, 727 triangles are generated (Figure 8) and mean area of them is 13544 $m^2$ (equivalent to 116 $m$ horizontal resolution).

6. Time-Series data processing, including the forcing data, LAI and melting factor. Also, the Thiessen Polygons of forcing sites are generated, which tell the matching TSD for each cell.

7. Attaching the attributes of soil, geology, and land cover to the triangular mesh.

8. Building the topological relationship between rivers and triangles.

9. Generating the model configuration files.

10. Writing the model input files out.

### 5.2.2 Result Visualization

Once the simulation is complete, we can analyze the results (Shu, 2020b). The script of visualization *demo_waerma_ana.R* still needs to repeat the first three steps of loading R libraries, setting up folders, and the environment.

The *shud.env* function configures the global variable environment, setting up several default variables for data processing to boost post-processing. The input arguments of *shud.env* include the project name, the path of model input, and model output.

After that, we start to load and plot the simulation results. A series of reading functions are available to read the model input files as well as the output files. The readout function reads the simulation results and returns multi-column time-series data, where the index for each column represents the index of HCUs. For example, the TSD for the $j$th river reach can be found in





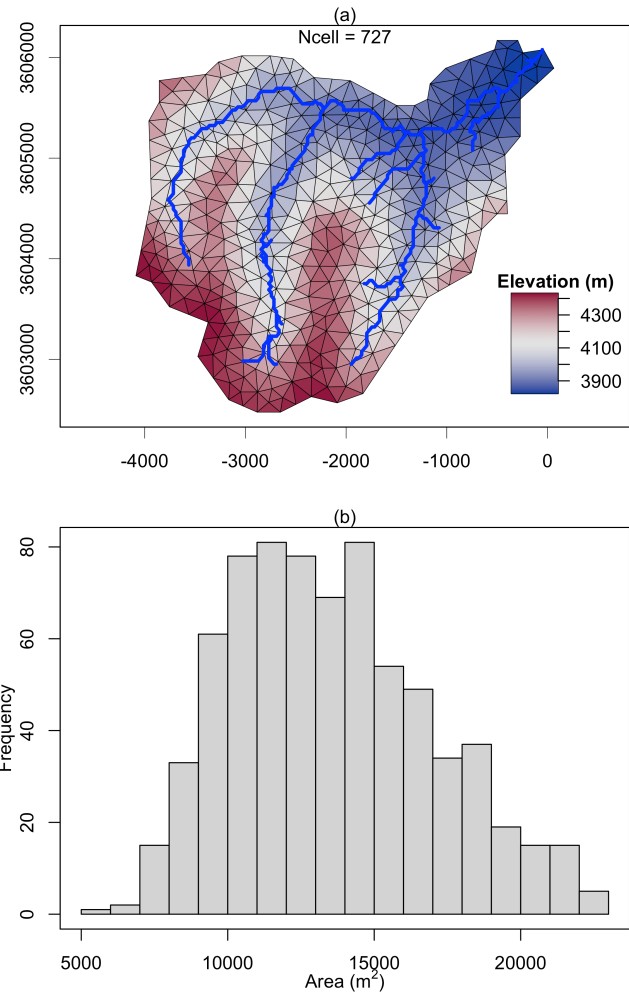

**Figure 7.** The triangular mesh (a) generated by the function *shud.triangle* for Waerma Watershed, and the histogram of triangles' area (b). The color in plot (a) is the centroid elevation of triangles.

column $j$ of stream flow data (.*rivqdown*) , whereas the TSD for the $j$th triangular cell can be found in column $j$ of the potential evapotranspiration file(.*elevetp*).

Figure 8 demonstrates the visualization of the hydrograph (precipitation versus discharge), water balance ( Storage change = Precipitation - Evapotranspiration - Discharge), the spatial distribution of groundwater table and annual mean evapotranspiration. Without calibration with observational data, this figure is outputs of preliminary simulation and the resulting values may not be effective. Since the script to read and plot results are self-explained, users can read and modify the code based on their own needs. The script of model deployment and result visualization demonstrates the capability of rSHUD for pre- and

post-processing of SHUD modeling.



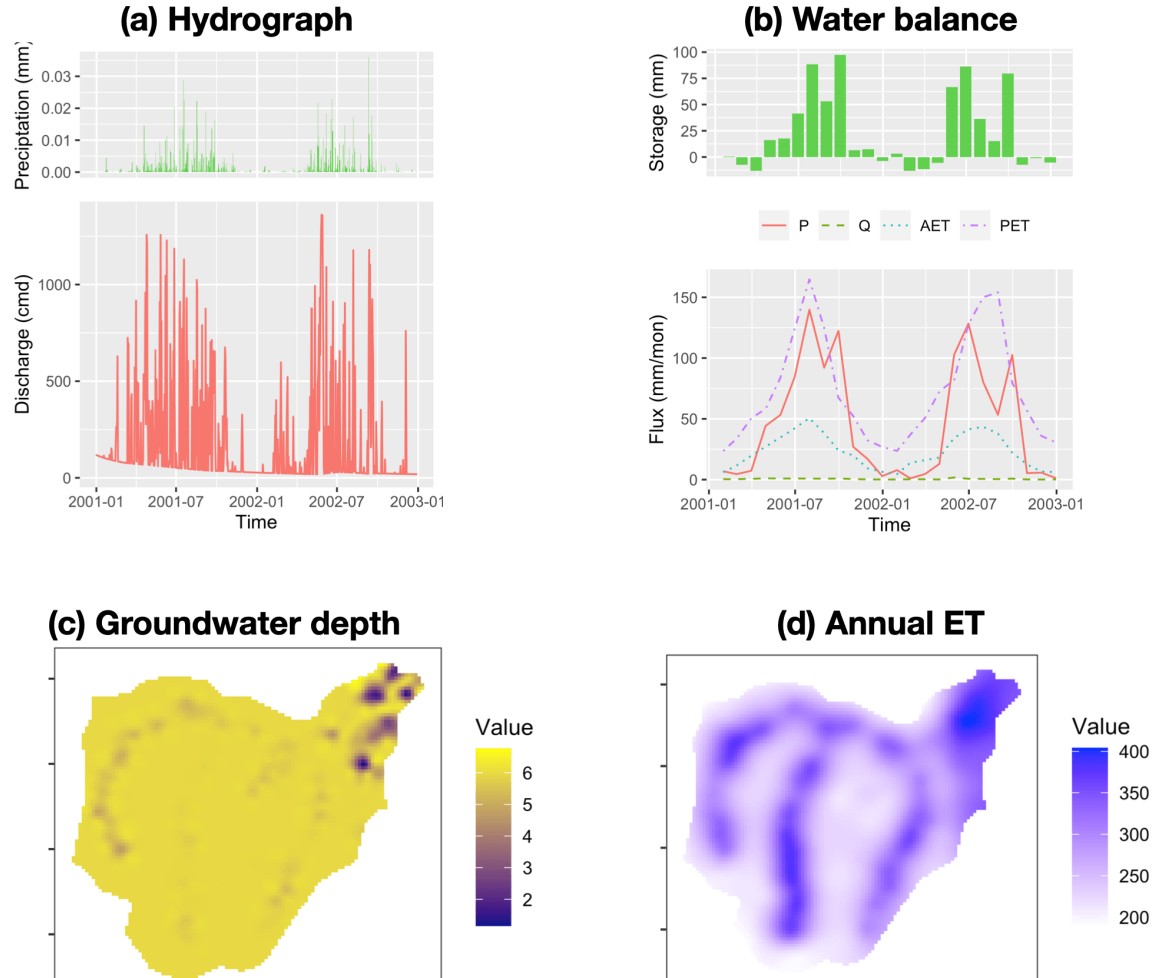

**Figure 8.** The result analysis of modeling Waerma Watershed; (a) hydrograph, (b) watershed scale water balance, (c) ground water depth $[m]$, and (d) annual evaporation rate $[mm/year]$.

## 6   Conclusion

The rSHUD is a toolbox developed in the R environment that supports the pre- and post-processing for the SHUD model.

The rSHUD package provides a set of tools to facilitate the conversion, parameterization, integration, analysis, and visualization of hydrologic data for the SHUD model. The package includes a toolkit for raster and vector data to construct unstructured triangular mesh domains. It also enables defining and adjusting hydraulic properties for soil types and land covers. The package ensures seamless integration with the SHUD model, with the ability to read and write input files and load output results. The package also enables detailed temporal and spatial analyses of hydrologic data and data visualization for easier interpretation.



The tools in rSHUD not only boost the model deployment and analysis for the SHUD model but also can be used for other spatial analysis and hydrological data processing. An automatic data processing and model deployment platform, Global Hydrological Data Cloud (https://shuddata.com, accessed June 2023), was implemented with the support of the rSHUD package.

*Code availability.*   The source code of the rSHUD model is kept updating at (https://github.com/SHUD-System/rSHUD, accessed June 2023) and uploaded to Zenodo (Shu, 2023a) . The help files are embedded in the rSHUD R package, user can use *help(command)* or *?command* to read the help page.

*Data availability.*   The data for building the Shallhills and Waerma Watershed model is embedded in the rSHUD package (Shu, 2023a) (https://github.com/SHUD-System/rSHUD, accessed June 2023). The model output of Waerma Watershed from SHUD model is achieved on Zenodo (Shu, 2020b)

*Sample availability.*   No sample is used in this study.

*Video supplement.*   No video is used in this study.

**Appendix A:  Land cover parameter table**

The table headers in this section have specific meanings as delineated below:

- INDEX: This is the index assigned to each row.

- ALBEDO: This refers to the land cover albedo represented as a dimensionless quantity.

- VEGFRAC: This parameter indicates the vegetation fraction and is expressed as a dimensionless quantity.

- ROUGH: This refers to the Manning roughness assigned to the land cover, expressed in units of $sm^{-1/3}$.

- RZD: This is the root depth of the vegetation and is expressed in units of $m$.

- SOILDGRD: This parameter indicates the soil degradation ratio, given as a dimensionless quantity.

- IMPAF: This parameter indicates the impervious fraction of the land cover, expressed as a dimensionless quantity.

- Classification: This refers to the name of the classification in its original datasets.





| INDEX | ALBEDO | VEGFRAC | ROUGH | RZD | SOILDGRD | IMPAF | Classification |
|---|---|---|---|---|---|---|---|
| 0 | 0.0700 | 0.0000 | 0.02 | 0.0000 | 0 | 0 | 0. Water / Goode's Interrupted Space |
| 1 | 0.0620 | 0.8000 | 0.06 | 1.0000 | 0 | 0 | 1. Evergreen Needleleaf Forest |
| 2 | 0.0760 | 0.9000 | 0.07 | 1.2500 | 0 | 0 | 2. Evergreen Broadleaf Forest |
| 3 | 0.0620 | 0.8000 | 0.06 | 1.0000 | 0 | 0 | 3. Deciduous Needleleaf Forest |
| 4 | 0.0920 | 0.8000 | 0.06 | 1.2500 | 0 | 0 | 4. Deciduous Broadleaf Forest |
| 5 | 0.0690 | 0.7950 | 0.06 | 1.1250 | 0 | 0 | 5. Mixed Cover |
| 6 | 0.0752 | 0.7999 | 0.05 | 0.9975 | 0 | 0 | 6. Woodland |
| 7 | 0.0908 | 0.8018 | 0.04 | 0.8721 | 0 | 0 | 7. Wooded Grassland |
| 8 | 0.0991 | 0.6250 | 0.05 | 0.6508 | 0 | 0 | 8. Closed Shrubland |
| 9 | 0.1213 | 0.2182 | 0.04 | 0.5777 | 0 | 0 | 9. Open Shrubland |
| 10 | 0.1073 | 0.7255 | 0.04 | 0.7500 | 0 | 0 | 10. Grassland |
| 11 | 0.1005 | 0.8354 | 0.04 | 0.7500 | 0.5 | 0 | 11. Cropland |
| 12 | 0.1595 | 0.0749 | 0.03 | 0.5500 | 0.6 | 0 | 12. Bare Ground |
| 13 | 0.0971 | 0.7436 | 0.02 | 0.7972 | 0.9 | 0.9 | 13. Urban and Built-Up |

**Table A1.** The parameters for the University of Maryland (UMD) Global Land Cover Classification (Hansen et al., 2000).





| INDEX | ALBEDO | VEGFRAC | ROUGH | RZD | SOILDGRD | IMPAF | Classification |
|---|---|---|---|---|---|---|---|
| 0 | 0.080 | 0.000 | 0.020 | 0.000 | 0.000 | 0.000 | Water |
| 1 | 0.140 | 0.800 | 0.070 | 1.0000 | 0.000 | 0.000 | Evergreen Needle leaf Forest |
| 2 | 0.100 | 0.900 | 0.070 | 1.2500 | 0.000 | 0.000 | Evergreen Broadleaf Forest |
| 3 | 0.140 | 0.800 | 0.070 | 1.0000 | 0.000 | 0.000 | Deciduous Needle leaf Forest |
| 4 | 0.120 | 0.800 | 0.070 | 1.2500 | 0.000 | 0.000 | Deciduous Broadleaf Forest |
| 5 | 0.110 | 0.700 | 0.060 | 1.1250 | 0.000 | 0.000 | Mixed Forests |
| 6 | 0.120 | 0.700 | 0.060 | 0.6508 | 0.000 | 0.000 | Closed Shrublands |
| 7 | 0.180 | 0.500 | 0.050 | 0.5777 | 0.000 | 0.000 | Open Shrublands |
| 8 | 0.100 | 0.625 | 0.045 | 0.9975 | 0.000 | 0.000 | Woody Savannas |
| 9 | 0.150 | 0.218 | 0.045 | 0.8721 | 0.000 | 0.000 | Savannas |
| 10 | 0.150 | 0.726 | 0.040 | 0.7500 | 0.000 | 0.000 | Grasslands |
| 11 | 0.100 | 0.200 | 0.035 | 0.600 | 0.000 | 0.000 | Permanent Wetland |
| 12 | 0.250 | 0.835 | 0.040 | 0.7500 | 0.500 | 0.000 | Croplands |
| 13 | 0.246 | 0.200 | 0.010 | 0.7972 | 0.900 | 0.900 | Urban and Built-Up |
| 14 | 0.200 | 0.835 | 0.040 | 0.7500 | 0.500 | 0.000 | Cropland/Natural Vegetation Mosaic |
| 15 | 0.650 | 0.000 | 0.020 | 0.000 | 0.000 | 0.500 | Snow and Ice |
| 16 | 0.300 | 0.010 | 0.035 | 0.5500 | 0.600 | 0.000 | Barren or Sparsely Vegetated |

**Table A2.** The parameters for USGS 0.5-km MODIS Global Land Cover (Broxton et al., 2014).

| INDEX | ALBEDO | VEGFRAC | ROUGH | RZD | SOILDGRD | IMPAF | Classification |
|---|---|---|---|---|---|---|---|
| 1 | 0.200 | 0.835 | 0.040 | 0.75 | 0.5 | 0.0 | Cropland |
| 2 | 0.150 | 0.800 | 0.070 | 1.00 | 0.0 | 0.0 | Forest |
| 3 | 0.150 | 0.600 | 0.060 | 0.65 | 0.0 | 0.0 | Shrub |
| 4 | 0.150 | 0.726 | 0.040 | 0.75 | 0.0 | 0.0 | Grassland |
| 5 | 0.080 | 0.000 | 0.020 | 0.00 | 0.0 | 0.0 | Water |
| 6 | 0.650 | 0.000 | 0.020 | 0.00 | 0.0 | 0.5 | Snow/Ice |
| 7 | 0.300 | 0.010 | 0.035 | 0.55 | 0.6 | 0.0 | Barren |
| 8 | 0.246 | 0.200 | 0.010 | 0.80 | 0.9 | 0.9 | Impervious |
| 9 | 0.100 | 0.200 | 0.035 | 0.60 | 0.0 | 0.0 | Wetland |

**Table A3.** The parameters for China Land Cover Dataset (Yang and Huang, 2021).

## A1   UMD land cover classification

## A2   MODIS Global Land Cover

## A3   China Land Cover Dataset

## Appendix B:  Pedotransfer function

$$K_{sat} = exp(7.755 + 0.03252 * p_s + 0.93 * t_p - 0.967 * \rho_b * \rho_b - 0.000484 * p_c * p_c$$

$$- 0.000322 * p_s * p_s + 0.001/p_s - 0.0748/p_{om} - 0.643 * log(p_s) - 0.01398 * \rho_b * p_c$$





Where:

- $K_{sat}$ - Saturated conductivity [$mday^{-1}$].

- $\theta$ - Porosity of soil/geology layer [$m^3 m^{-3}$].

- $\alpha$ - Coefficient in van Genutchen equation [$m^{-1}$].

- $\beta$ - Coefficient in van Genutchen equation [$-$].

- $p_s$ - Weight percentage of silt in soil [%].

- $p_c$ - Weight percentage of clay in soil [%].

- $p_{om}$ - Weight percentage of organic matter in soil [%].

- $\rho_b$ - Bulk density of soil [%].

- $t_p$ - Flag indicating the top/bottom layer. $t_p = 0$, top layer; $t_p = 1$, bottom layer.

## Appendix C: R script

### C1   Model deployment, Shale Hill CZO

```
rm(list=ls())
     clib=c('rgdal', 'rgeos', 'raster', 'sp', 'fields')
     x=lapply(clib, library, character.only=T)

     library(rSHUD)
prjname = 'sh'
     model.in <- file.path('../demo/input', prjname)
     model.out <- file.path('../demo/output', paste0(prjname, '.out'))
     fin=shud.filein(prjname, inpath = model.in, outpath = model.out )

dir.create(model.in, showWarnings = F, recursive = T)

     load("./data/sh.rda")
     wbd=sh[['wbd']]
     riv=sh[['riv']]
dem=sh[['dem']]
     tsd.forc=sh[['forc']]
```





```
     # sl = terrain(dem, v=slope, unit='tangent')
     # cellStats(sl, quantile)
     a.max = 200;
     q.min = 33;
     tol.riv = 5
     tol.wb = 5
aqd=3
     NX = 800
     years = seq(as.numeric(format(min(time(tsd.forc)), '%Y')),
                 as.numeric(format(max(time(tsd.forc)), '%Y')))
     ndays = days_in_year(years)

     riv.simp = rgeos::gSimplify(riv, tol=tol.riv, topologyPreserve = T)
     riv.simp = sp.CutSptialLines(sl=riv.simp, tol=20)

wb.dis = rgeos::gUnionCascaded(wbd)
     wb.simp = rgeos::gSimplify(wb.dis, tol=tol.wb, topologyPreserve = T)

     # shp.riv =raster::crop(riv.simp, wb.simp)
     # shp.wb = raster::intersect( wb.simp, riv.simp)
     tri = shud.triangle(wb=wb.simp,q=q.min, a=a.max, S=NX)
     # generate  .sp.mesh
     pm=shud.mesh(tri,dem=dem, AqDepth = aqd)
     sp.mesh=sp.mesh2Shape(pm=pm)
ncell = nrow(pm@mesh)
     print(ncell)
     # generate  .sp.att

     pa=shud.att(tri)
     # generate  .riv
```



```
    pr=shud.river(riv.simp, dem = dem)

    # Cut the rivers with triangles
spm = sp.mesh2Shape(pm)
    crs(spm) =crs(riv)
    spr=riv
    sp.seg = sp.RiverSeg(spm, spr)
    # Generate the River segments table
prs = shud.rivseg(sp.seg)

    # Generate initial condition
    pic = shud.ic(nrow(pm@mesh), nrow(pr@river), AqD = aqd, p1 = 0.2, p2=0.2)

go.plot <- function(){
      z=getElevation(pm = pm)
      loc = getCentroid(pm=pm)
      idx.ord = order(z)
      col=colorspace::diverge_hcl(length(z))
plot(sp.mesh[idx.ord, ], axes=TRUE, col=col, lwd=.5); plot(spr , col='blue', add=T, lwd=3)
      # image.plot( legend.only=TRUE, zlim= range(z), col=col, horizontal = T,legend.lab="Elevat
      #              smallplot= c(.6,.9, 0.22,0.26))
      image.plot( legend.only=TRUE, zlim= range(z), col=col, horizontal = F,
                  legend.args = list('text'='Elevation (m)', side=3, line=.05, font=2, adj= .2),
smallplot= c(.79,.86, 0.20,0.4))
    }
    ia = getArea(pm=pm)
    png(filename = '~/sh_mesh.png', height = 9, width = 6, res = 400, units = 'in')
    par(mfrow=c(2,1), mar=c(3, 3.5, 1.5,1) )
go.plot();
    mtext(side=3, text = '(a)')
    hist(ia, xlab='', nclass=20, main='', ylab='')
    mtext(side=3, text = '(b)')
    mtext(side=2, text = 'Frequency', line=2)
mtext(side=1, text = expression(paste("Area (", m^2, ")")), line=2)
    box();
```





```
# grid()
dev.off()

sp.c = SpatialPointsDataFrame(gCentroid(wb.simp, byid = TRUE),
                              data=data.frame('ID' = 'forcing'), match.ID = FALSE)
sp.forc = ForcingCoverage(sp.meteoSite = sp.c,  pcs=crs(wb.simp), dem=dem, wbd=wbd)
write.forc(sp.forc@data,
           path = file.path('./input', prjname),
           startdate = format(min(time(tsd.forc)), '%Y%m%d'),
           file=fin['md.forc'])
write.tsd(tsd.forc, file = file.path(fin['inpath'], 'forcing.csv'))

# model configuration, parameter
cfg.para = shud.para(nday=ndays)
# calibration
cfg.calib = shud.calib()

para.lc = lc.NLCD(lc=42) # 42 is the forest in NLCD classes
para.soil = PTF.soil()
para.geol = PTF.geol()

tsd.lai =  LaiRf.NLCD(lc=42, years=years)
write.tsd(tsd.lai$LAI, file = fin['md.lai'])

tsd.mf = MeltFactor(years=years)
write.tsd(tsd.mf, file = fin['md.mf'])
# write  input files.
write.mesh(pm, file = fin['md.mesh'])
write.riv(pr, file=fin['md.riv'])
write.ic(pic, file=fin['md.ic'])

write.df(pa, file=fin['md.att'])
write.df(prs, file=fin['md.rivseg'])
write.config(cfg.para, fin['md.para'])
```





```
      write.config(cfg.calib, fin['md.calib'])

      write.df(para.lc, file=fin['md.lc'])
write.df(para.soil, file=fin['md.soil'])
      write.df(para.geol, file=fin['md.geol'])
      writeshape(riv.simp, file=file.path(dirname(fin['md.att']), 'riv'))
      print(nrow(pm@mesh))
```

**C2   Model deployment, Waerma watershed**

```
rm(list=ls())
      # === 1. load library ============
      clib=c('rgdal', 'rgeos', 'raster', 'sp', 'fields', 'xts')
      x=lapply(clib, library, character.only=T)
      library(rSHUD)

      # === 2. create directories  ============
      dir.prj = '~/Documents/Ex_waerma'
      dir.forc = file.path(dir.prj, 'forc')
      dir.fig = file.path(dir.prj, 'figure')
dir.create(dir.forc, showWarnings = FALSE, recursive = TRUE)
      dir.create(dir.fig, showWarnings = FALSE, recursive = TRUE)

      # === 3. setup the project ============
      prjname = 'waerma'
model.in <- file.path(dir.prj, 'input', prjname)
      model.out <- file.path(dir.prj, 'output', paste0(prjname, '.out'))
      fin=shud.filein(prjname, inpath = model.in, outpath = model.out )
      if (dir.exists(model.in)){
        unlink(model.in, recursive = T, force = T)
}
      dir.create(model.in, showWarnings = F, recursive = T)

      # === 4. load and reproject data ============
      data(waerma)
wbd=waerma[['wbd']]
```





```
meteosite = waerma[['meteosite']] # This is in GCS

crs.pcs = crs.Albers(wbd)
dem = projectRaster(waerma[['dem']], crs=crs.pcs)
wbd= spTransform(waerma[['wbd']], CRSobj = crs.pcs)
riv= spTransform(waerma[['riv']], CRSobj = crs.pcs)

# sl=mask(terra::terrain(dem, opt='slope', unit='tangent'), wbd)
# plot(sl)

r0.soil = waerma[['soil']]
att.soil = waerma[['att']]$soil
rcl.soil=cbind(att.soil[, 1], 1:nrow(att.soil))
r.soil = projectRaster(reclassify(r0.soil, rcl.soil), crs = crs.pcs, method="ngb")

r0.geol = waerma[['geol']]
att.geol = waerma[['att']]$geol
rcl.geol=cbind(att.geol[, 1], 1:nrow(att.geol))
r.geol = projectRaster(reclassify(r0.geol, rcl.geol), crs = crs.pcs, method="ngb")

r0.lc = waerma[['lc']]
att.lc = waerma[['att']]$lc
rcl.lc=cbind(att.lc[, 1], 1:nrow(att.lc))
r.lc = projectRaster(reclassify(r0.lc, rcl.lc), crs = crs.pcs, method="ngb")

# === 5. some threshold for model deployment ============
AREA = 9853260 # KNOWN Area
a.max = 150*150;
q.min = 33;
tol.riv = 50
tol.wb = 50
aqd = 6
NX = AREA / a.max
ndays = 731
```





```
# === 6. domain decomposition ============
# simplify the river network.
riv.simp = rgeos::gSimplify(riv, tol=tol.riv, topologyPreserve = T)

# desolve and simplify the watershed boundary
wb.dis = rgeos::gUnionCascaded(wbd)
wb.simp = rgeos::gSimplify(wb.dis, tol=tol.wb, topologyPreserve = T)

# !! Triangulation
tri = shud.triangle(wb=wb.simp,q=q.min, a=a.max, S=NX)
# generate  .sp.mesh
pm=shud.mesh(tri,dem=dem, AqDepth = aqd)
sp.mesh=sp.mesh2Shape(pm=pm)
ncell = nrow(pm@mesh)
print(ncell)

# generate  .riv
pr=shud.river(riv.simp, dem = dem)


# === 7. TSD DATA ============
fns.meteo  =  paste0(meteosite$FILENAME, '.csv')
tsd.forc = waerma$tsd$forc
range(time(tsd.forc[[1]]))
for(i in 1:length(fns.meteo)){
   write.tsd(tsd.forc[[i]], file = file.path(dir.forc, fns.meteo[i]))
}
tsd.mf = MeltFactor(years = seq(as.numeric(format(min(time(tsd.forc[[1]])), '%Y')),
                               as.numeric(format(max(time(tsd.forc[[1]])), '%Y'))))
tsd.lai = waerma$tsd$lai[[1]]
# Coverage of meteorological sites.
sp.forc = ForcingCoverage(sp.meteoSite = meteosite,
                              filenames= fns.meteo,
                              pcs=crs.pcs, gcs=crs(meteosite),
715                              dem=dem, wbd=wbd)
```





```
write.forc(sp.forc@data, path = normalizePath(dir.forc),
          startdate = format(min(time(tsd.forc[[1]])), '%Y%m%d'),
          file=fin['md.forc'])

# === 8. attributes ============
     # generate  .sp.att
     pa=shud.att(tri, r.soil = r.soil, r.geol = r.geol, r.lc=r.lc, r.forc = sp.forc)
     head(pa)

# === 9. toplogical relation between river and triangle  ============
     # Cut the rivers with triangles
     spm = sp.mesh2Shape(pm)
     crs(spm) =crs(riv)
     spr=riv
sp.seg = sp.RiverSeg(spm, spr)
     # Generate the River segments table
     prs = shud.rivseg(sp.seg)

     # Generate initial condition
pic = shud.ic(nrow(pm@mesh), nrow(pr@river), AqD = aqd)

     go.plot <- function(){
       z=getElevation(pm = pm)
       loc = getCentroid(pm=pm)
idx.ord = order(z)
       col=colorspace::diverge_hcl(length(z))
       plot(sp.mesh[idx.ord, ], axes=TRUE, col=col, lwd=.5); plot(spr , col='blue', add=T, lwd=3)
       image.plot( legend.only=TRUE, zlim= range(z), col=col, horizontal = F,
                  legend.args = list('text'='Elevation (m)', side=3, line=.05, font=2, adj= .2),
smallplot= c(.79,.86, 0.20,0.4))
     }
     ia = getArea(pm=pm)
     png(filename = file.path(dir.fig, paste0(prjname, '_mesh.png')), height = 9, width = 6, res
     par(mfrow=c(2,1), mar=c(3, 3.5, 1.5,1) )
go.plot();
```





```
      mtext(side=3, text = '(a)')

      mtext(side=3, text = paste0('Ncell = ', ncell), line=-1)

      hist(ia, xlab='', nclass=20, main='', ylab='')

      mtext(side=3, text = '(b)')

mtext(side=2, text = 'Frequency', line=2)

      mtext(side=1, text = expression(paste("Area (", m^2, ")")), line=2)

      box();

      # grid()

      dev.off()


      # === 10. configuration files  ============

      # model configuration, parameter

      cfg.para = shud.para(nday=ndays)

# calibration file

      cfg.calib = shud.calib()

      para.lc = lc.GLC()

      para.soil = PTF.soil(att.soil[, -1])   # only 4 columns (Silt, clay, OM, bulk density) as inp

para.geol = PTF.geol(att.geol[, -1])

      # === 11. write  input files. ============

      write.mesh(pm, file = fin['md.mesh'])

      write.riv(pr, file = fin['md.riv'])

write.ic(pic, file = fin['md.ic'])

      write.df(pa, file=fin['md.att'])

      write.df(prs, file=fin['md.rivseg'])

      write.config(cfg.para, fin['md.para'])

write.config(cfg.calib, fin['md.calib'])

      write.tsd(tsd.lai, fin['md.lai'] )

      write.tsd(tsd.mf, fin['md.mf'] )

write.df(para.lc, file=fin['md.lc'])
```





```
write.df(para.soil, file=fin['md.soil'])
write.df(para.geol, file=fin['md.geol'])
writeshape(riv.simp, file=file.path(dirname(fin['md.att']), 'riv'))
print(nrow(pm@mesh))
```

**C3   Post-processing, Waerma watershed**

```
rm(list=ls())

# === pre1. load library ============
clib=c('rgdal', 'rgeos', 'raster', 'sp', 'fields', 'xts', 'ggplot2')
x=lapply(clib, library, character.only=T)
library(rSHUD)

# === pre2. create directories  ============
dir.prj = '~/Documents/Ex_waerma'
dir.forc = file.path(dir.prj, 'forc')
dir.fig = file.path(dir.prj, 'figure')
dir.create(dir.forc, showWarnings = FALSE, recursive = TRUE)
dir.create(dir.fig, showWarnings = FALSE, recursive = TRUE)

# === pre3. setup the project ============
prjname = 'waerma'
model.in <- file.path(dir.prj, 'input', prjname)
# model.out <- file.path(dir.prj, 'output', paste0(prjname, '.out'))
model.out <- '~/Documents/output/waerma.out'
fin=shud.filein(prjname, inpath = model.in, outpath = model.out )
shud.env(prjname, inpath = model.in, outpath = model.out )
dir.create(model.in, showWarnings = F, recursive = T)

ia=getArea()
ncell=length(ia)
spm=sp.mesh2Shape()

gplotfun <- function(r, leg.lab='value'){
```





```
map.p <- rasterToPoints(r)
       #Make the points a dataframe for ggplot
       df <- data.frame(map.p)
       #Make appropriate column headings
       colnames(df) <- c('X', 'Y', 'Value')
#Now make the map
       g= ggplot(data=df, aes(y=Y, x=X)) +
         geom_raster(aes(fill=Value)) +
         # geom_point(data=sites, aes(x=x, y=y), color="white", size=3, shape=4) +
         theme_bw() + coord_equal() +
# scale_fill_continuous(leg.lab) +
         theme(
           # axis.title.x = element_text(size=16),
           # axis.title.y = element_text(size=16, angle=90),
           # axis.text.x = element_text(size=14),
# axis.text.y = element_text(size=14),
           axis.title.x = element_blank(),
           axis.title.y = element_blank(),
           axis.text.x = element_blank(),
           axis.text.y = element_blank(),
panel.grid.major = element_blank(),
           panel.grid.minor = element_blank(),
           legend.position = 'right',
           legend.key = element_blank()
         )
return(g)
     }
     gl=list()
     # === 1. plot Q (discharge) data ============
     oid = getOutlets()
qdown = readout('rivqdown')
     prcp = readout('elevprcp')
     xt = 1:(365*2)+365*1
     q=qdown[xt, oid]
     pq = cbind(q, rowMeans(prcp[xt,]))[,2:1]
```





```
# gl[[1]] = autoplot(q)+xlab('')+ylab('Discharge (m^3/day)')+theme_bw()
      gl[[1]] = hydrograph(pq, ylabs = c('Preciptation (mm)', 'Discharge (cmd)'))
      gl[[1]]

      # === 2. plot Water Balance ============
xl=loaddata(varname=c('rivqdown', 'eleveta', 'elevetp', 'elevprcp', 'eleygw'))
      wb=wb.all(xl=xl, plot=F)[(1:24)+12, ]*1000
      gl[[2]] = hydrograph(wb, ylabs = c('Storage (mm)', 'Flux (mm/mon)'), legend.position='top')
      gl[[2]]

# === 3. plot groundwater data ============
      eleygw = readout('eleygw')[xt, ]
      ts.gw=apply.daily(eleygw, sum)/ncell
      # plot(ts.gw)
      gw.mean = apply(eleygw, 2, mean)
aqd =getAquiferDepth()
      r.gw = MeshData2Raster(gw.mean)
      d.gw = aqd - r.gw
      d.gw[d.gw<0]=0
      gl[[3]] =gplotfun(d.gw, leg.lab='Depth (m)')+
scale_fill_gradient(low = "darkblue", high = "yellow")
      gl[[3]]

      # === 4. plot ETa data ============
      eleveta = readout('eleveta')[xt, ]
ts.eta=apply.monthly(eleveta, sum)
      # plot(ts.eta)
      eta.mean = apply(eleveta, 2, mean)*365
      r.eta = MeshData2Raster(eta.mean)*1000 # mm/day
      # plot(spm, axes=TRUE)
# plot(add=T, r.eta)
      gl[[4]]=gplotfun(r.eta, leg.lab='Rate (mm/day) ')+
        scale_fill_gradient(low = "white", high = "blue")
      gl[[4]]
```





```
# === Saving the plots ============
     gg=gridExtra::arrangeGrob(grobs=gl, nrow=2, ncol=2)
     ggsave(plot = gg, filename = file.path(dir.fig, 'waerma_res.png'), width = 7, height=7, dpi=

     for(i in 1:4){
ggsave(plot = gl[[i]], filename = file.path(dir.fig, paste0('waerma_res_', i, '.png')),
            width = 3.5, height=4, dpi=400, units = 'in')
     }
```

| package | Version | Use in rSHUD |
|---|---|---|
| *Rcpp* | 1.0 | To support the C/C++ program. |
| *reshape2* | 1.4 | Convert any data to data.frame type. |
| *ggplot2* | 3.4 | Powerful plot functions. |
| *gridExtra* | 2.3 | Plot functions. |
| *grid* | 4.2 | Graphics package |
| *fields* | 10.3 | Tools for spatial data |
| *xts* | 0.10 | Tools for time series data. |
| *hydroGOF* | 0.4 | To calculate the goodness of fitting. |
| *zoo* | 1.8 | Tools for time series data. |
| *raster* | 3.5 | Raster data modeling and analysis. |
| *sp* | 1.5 | Vector data modeling and analysis. |
| *rgeos* | 0.5 | Geoanalysis tools. |
| *RTriangle* | 1.6 | Jonathan Shewchuk's Triangle library |
| *rgdal* | 1.5 | Geospatial data library |
| *proj4* | 1.0 | PROJ.4 cartographic projections library |
| *abind* | 1.0 | Combine multidimensional arrays |
| *utils* | 4.2 | R Utils Package |
| *lubridate* | 1.9 | Handle the date-time data. |
| *geometry* | 0.4 | Mesh Generation and Surface Tessellation |
| *methods* | 4.2 | To heritage functions. |
| *ncdf4* | 1.19 | To support the NetCDF data. |
| *GGally* | 2.1 | Extension of ggplot2. |
| *doParallel* | 1.0 | Parallel computing. |

**Table A1.** R packages required in rSHUD and their functionalities.



*Author contributions.* Lele Shu – Conceptualization, Investigation, Methodology, Software, Validation, Visualization, Writing original draft and editing

Paul Ullrich – Supervision, Investigation, Writing original draft and editing

Xianhong Meng – Supervision, Investigation, Writing original draft and editing

Christopher Duffy – Supervision, Investigation, Writing original draft and editing

Hao Chen – Code development, program test, writing and editing.

Zhaoguo Li– Code development, program test, writing and editing.

*Competing interests.* Lele Shu and Paul Ullrich are members of the editorial board of the journal

*Acknowledgements.* Authors Shu was supported by the Chinese Academy Sciences (E0290304, xbzg-zdsys-202215), National Cryosphere Desert Data Center (E01Z790215), the Qinghai Key Laboratory of Disaster Prevention (QFZ-2021-Z02) and the California Energy Commission grant (award no. EPC-16-063).

Co-author Meng was supported by the National Natural Science Foundation of China (41930759).

Co-author Ullrich was supported by the U.S. Department of Energy Regional and Global Climate Modeling Program (RGCM) "An Integrated Evaluation of the Simulated Hydroclimate System of the Continental US" project (award no. DE-SC0016605) and the National Institute of Food and Agriculture, U.S. Department of Agriculture, California Agricultural Experiment Station hatch project accession no. 1016611.





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
