# Peer review of "rSHUD v2.0: Advancing SHUD and Unstructured Hydrological Modeling in the R Environment"

_Geoscientific Model Development, 2023_

## Referee Comment (RC1)

**Review of "rSHUD v2.0: Advancing Unstructured Hydrological Modeling in the R Environment" by Shu et al.**

The basic idea behind the paper, that is, availability of an easy-to-use modeling package is good. The rSHUD package should be useful for easier and convenient use of the model. However, the important question is why someone should choose the SHUD model. There are many other distributed hydrological models such as VIC, tRIBS, and MIKE-SHE which are doing something very similar to what the SHUD model is doing. Also, these models have a long legacy. In this case, how does a hydrologist decide upon which model should be used? Perhaps, the better question is what are the advantages of the SHUD model over the other already existing and well tested models? This information needs to be discussed in the paper.

The description of the model is quite unclear. Does the model use 1-D or 2-D shallow water equation for surface runoff? Is there any provision for capillary fringe above the saturated zone in subsurface? Does the model simulate both the Hortonian and Dunnian overland flow? How is the infiltration calculated, using Richards equation? How is potential ET and, subsequently, actual ET calculated? I think it is important to provide this information in the paper since a hydrologist would want to know what does this package offer that the other packages do not? In fact, the most difficult part of running a giant hydrological model such as SHUD is that these are somewhat opaque to a beginner and the conceptual comparisons with other existing models are difficult.

The model estimates soil hydraulic parameters using pedotransfer functions (PTFs)? Are the PTFs used by the rSHUD package universal in some sense? Also, would the PTFs be able to account for macropores which can be very important for percolation? This brings another crucial point: Most of the effort in hydrological modeling is in calibration of the model parameters. This issue has not even been discussed in the paper. Then, there are uncertainties associated with the modeling and hydrological data which are known to the hydrological community for the last 40 years. Should not a package like rSHUD which aims to make data pre-processing easier provide a program for accounting of uncertainties in data?

Why are the observed streamflows not shown in Figure 8. In fact, the simulated streamflow does not seem very realistic to me; look at the pronounced recession? It seems like a consequence of using unrealistic parameters for the simulation. Therefore, the question is if the rSHUD package provides any tool for model parameter adjustment (calibration)? This is a very important question given that most of the time in modeling is spent in the calibration, not preprocessing or postprocessing of data? The latter is important as the authors point out, but not the main problem in my mind. Further, the monthly Q values are zero for all the months in Figure 8 which does not make sense given that daily values are pretty significant. Why are the result for SHCZO basin not shown?

The way the paper is written, it is unclear at many places what parts of the modeling process are automated and what parts the user needs to take care of?

**Specific comments:**

Line 47: mathematical equations

Line 49: Also, for hypothesis testing

Line 61-64: Should completely automated modeling even be the aim of process-based modeling given that we want to understand our watershed's functioning which is not possible by ML/DL methods? Note that DL models are still based on correlations and as such do not yield much understanding about physics. Also, given the uncertainties in data which are dominantly epistemic, a fully automated procedure does not even seem like a scientifically sound goal. So, should not the rSHUD package contain some program for data pre-processing so that the user is able to take care of these epistemic uncertainties?

Line 132: solver

Line 146: Really? I am not sure what you mean by 'river outlets at the edge of the watershed'?

Line 161: What slope? Slope of the phreatic surface or the bedrock?

Line 171: Combination of functions

Line 185: What is mean by 'reference data'?

Line 207: Is user supposed to standardize these files? Any guidance on how to do it?

Line 215: "It is necessary that … can occur otherwise".

Line 241: What about the depth to bedrock?

Line 251: 'the' instead of 'The'

Line 275: Any references for this?

Line 285: Explain how the hydraulic parameters are generated.

Line 288: What do you mean by 'overlapping relationship'?

Line 300: a cell

Line 308: 'Vertical flux of water from the surface to the soil' is infiltration? Why not just write infiltration?

Also, adding a table of all the model parameters would be really helpful for the readers.

Line 317-324: Clarify what the user needs to do to tackle this issue?

Line 327: reads

Line 332: What is 'experiential'? So, the melting factor stays the same for each day in a month? Would this not introduce any artifacts at the end of the month where the melting factor suddenly changes?

Equation 1: Subscripts in the equation should not be italic as these are not variables. Same comments apply for equation in the Appendix B.

Line 341: But a modeler would always want to and should use the observational data to constrain the model if such data are available? Your results in Figure 8 show that the parameters used are not adequate.

Why are the results for SCHZO not shown?

Line 406: What do you mean by 'expected minimum resolution of the modeling'?

Line 409: Thiessen polygons are generated for interpolating meteorological data, right? If so, mention this. Also, I am assuming that the meteorological data are used in distributed manner by the model, right?

Line 439: Not clear what you mean by other spatial analysis and hydrological data processing?

Line 445: archived

---

## Author Comment (AC1)

**OVERVIEW**

The study describes an open-source toolkit in R to facilitated the pre- and post-processing steps in using the SHUD hydrological model.

**GENERAL COMMENTS**

The paper is well written and clear. The topic is relevant as the toolkit in R is expected to facilitate the users of SHUD model. However, I have one major comment that needs to be addressed.

Reading the title, I expected to read about the development of an open system tool to apply multiple hydrological modelling approaches, with the possibility to consider different configurations of the processes to be simulated (title: "Advancing Unstructured Hydrological Modelling"). However, the paper describes a toolkit to facilitate the use of a single model, SHUD. That's fine, but it needs to be made clear in the introduction and in the title. Otherwise the reader would expect a different content.

Thank you for your feedback. We appreciate your perspective on the title and the content of the paper. Our intention with the title "Advancing Unstructured Hydrological Modeling" was to emphasize the broader applicability of the rSHUD toolkit beyond just the SHUD model. While the primary focus of the paper is on the SHUD model, many functions within the rSHUD toolkit are designed to be versatile and can be applied to other similar hydrological models.

For instance, functions like MeshData2Raster() can convert values on an unstructured triangular mesh into a regular grid raster through spatial interpolation. Similarly, the shud.triangle() function, which generates the geometry of an unstructured triangular mesh from a Shapefile Polygon, is not exclusively tailored for the SHUD model. In fact, out of the 160+ functions in rSHUD, a majority are not strictly limited to SHUD and can be utilized for modeling and analysis tasks in other similar hydrological models.

However, in light of your feedback, we recognize the potential for confusion and consider revising the title and introduction to more accurately reflect the content and scope of the paper. Now we change the title as "rSHUD v2.0: Advancing SHUD and Unstructured Hydrological Modeling in the R Environment".

**SPECIFIC COMMENTS (L: line or lines)**

**L53: GUI, acronyms should be defined.**

Thank you.

We updated the text in revise manuscript. GUI is Graphical User Interface.

**L57: Exactly the same in this paper, a toolkit developed for just one model. As mentioned above, I would suggest changing the introduction.**

Thank you for your feedback. We acknowledge the concern raised. While the primary focus of the rSHUD toolkit is to support the SHUD model, its functionalities extend beyond just this model. However, to ensure clarity and avoid potential confusion, we revised the introduction to more accurately reflect the scope and capabilities of the rSHUD toolkit. Your insights are invaluable, and we appreciate your guidance in this matter.

**L77-88: I believe it is not needed to introduce R in the paper.**

Thank you for your feedback.

We included a brief introduction to R to cater to readers who might be less familiar with it. However, recognizing that many in our target audience might already be well-acquainted with R, we revised this section to be more concise. We aim to strike a balance between providing context and ensuring the content remains directly relevant to the primary focus of the paper.

**L93: "The rSHUD version matches the SHUD model version". Which version? Where is it implemented? Which programming language?**

Thank you for pointing that out. In the revised manuscript, we will clarify as follows:

"The current version of rSHUD is 2.0, designed to support SHUD v2.0. To ensure compatibility and streamline user experience, The development team maintains concurrent versioning for both rSHUD and SHUD. While rSHUD is developed using the R programming language, SHUD is implemented in C/C++. The versioning process is managed manually to ensure consistency between the two."

**L95-96: Two lines of code for what? To be defined.**

Thank you for pointing it out. The two lines of code mentioned are intended to facilitate the installation of the rSHUD package and its dependent libraries in a fresh R environment. This allows new users to easily set up and start using the rSHUD package. We have now added this clarification in the manuscript for better context.

**L98: Table A1 does not contain the libraries, please check.**

Thank you for pointing it out. Due to formatting issues, Table A1 was mistakenly placed on page 36. We have now rectified this oversight, and Table A1 is correctly positioned in Appendix A.

**L129: "coupled with other systems such as …". To be removed or clarified. How can it be coupled? Add details.**

Thank you.  Upon reflection, the phrase "coupled with other systems such as …" does introduce ambiguity without adding substantive information to the context. To maintain clarity and conciseness, this phrase has been removed from line L129 as recommended.

**L200-204: This part is in first person "We". Different to everything else in the text, please revise.**

We revise the paragraph as:

"Multiple data processing stages were involved in this step. Holes were removed, modeling boundaries were projected, and buffer zones were generated in sequential procedures. Irrelevant data from the Digital Elevation Model (DEM) were excluded, retaining only pertinent information within the study area. The DEM data underwent reprojection and simplification into a Projected Coordinate System to facilitate analysis. The river flow direction consistency for the river network data was verified and corrected, while duplicate points and segments were eliminated, and the data format was standardized."

**L371: 59.4 mm of evapotranspiration seems to low, and the water balance in the basin is not closed. Please check.**

Thank you for pointing that out.

The data originally cited was sourced from the Shale Hill introduction of the Critical Zone Observatory website (https://czo-archive.criticalzone.org/shale-hills/infrastructure/field-area/susquehanna-shale-hills-critical-zone-observatory/). Upon further review, the value may not accurately represent the evapotranspiration for the region. Consequently, it has been replaced with data from more recent literature, including studies by Jin et al. (2011), Shi et al. (2013), and Brantley et al. (2018).

According to these references, the total annual precipitation is approximately 1000 mm/yr. The annual evapotranspiration ranges between 500-600 mm/yr, and the annual runoff is estimated to be between 400-600 mm/yr. This suggests that runoff constitutes about half of the total precipitation.

Shi, Y., Davis, K. J., Duffy, C. J., & Yu, X. (2013). Development of a Coupled Land Surface Hydrologic Model and Evaluation at a Critical Zone Observatory. Journal of Hydrometeorology, 14(5), 1401–1420. https://doi.org/10.1175/JHM-D-12-0145.1

Jin, L., Andrews, D. M., Holmes, G. H., Lin, H., & Brantley, S. L. (2011). Opening the "Black Box": Water Chemistry Reveals Hydrological Controls on Weathering in the Susquehanna Shale Hills Critical Zone Observatory. Vadose Zone Journal, 10(3), 928–942. https://doi.org/10.2136/vzj2010.0133

Brantley, S. L., White, T., West, N., Williams, J. Z., Forsythe, B., Shapich, D., et al. (2018). Susquehanna Shale Hills Critical Zone Observatory: Shale Hills in the Context of Shaver's Creek Watershed. Vadose Zone Journal, 17(1), 1–19. https://doi.org/10.2136/vzj2018.04.0092

**RECOMMENDATION**

**On this basis, I found the topic of the paper relevant, and I suggest a moderate revision before the paper can be published in GMD.**

Thank you for your feedback. Your insights are invaluable and we hope our revisions address your concerns.

---

## Author Comment (AC2)

Following are my reply to the comments. The reviewer's comments and questions are in bold, and my reply is in normal text.

**The basic idea behind the paper, that is, availability of an easy-to-use modeling package is good. The rSHUD package should be useful for easier and convenient use of the model. However, the important question is why someone should choose the SHUD model. There are many other distributed hydrological models such as VIC, tRIBS, and MIKE-SHE which are doing something very similar to what the SHUD model is doing. Also, these models have a long legacy. In this case, how does a hydrologist decide upon which model should be used? Perhaps, the better question is what are the advantages of the SHUD model over the other already existing and well tested models? This information needs to be discussed in the paper.**

Thank you for your insightful comments and for recognizing the potential utility of the rSHUD package. We acknowledge the importance of the question you raised regarding the selection of the SHUD model over other established hydrological models.

Your question, "why someone should choose the new model," is indeed a crucial one. Every emerging model faces this challenge, and the answer often emerges from continuous testing and scrutiny by the community over time. However, the intent of this paper is not necessarily to answer this question directly. The decision to adopt the SHUD model ultimately lies with the users, and the role of rSHUD is to facilitate this process by reducing the learning curve for new users and critics. Our primary aim is to present the unique features of the SHUD model and pave the way for its application and testing.

The SHUD Model description paper (Shu et al., 2020) has already elaborated on the model's design philosophy, computational methods, validation cases, capability demonstrations, and even future development plans. Comparing models is a complex and often controversial task. Evaluations encompass various aspects, including a

model's conceptual understanding, structure, algorithms, computational efficiency, application scenarios, and user-friendliness. Finding a universally accepted answer in this realm is challenging. Numerous scholars have delved into the intricacies of hydrological models, and while there's no definitive conclusion on which model is superior, there's a consensus that the abundance of models indicates our limited understanding of hydrological processes at the catchment scale.

In the revised manuscript, we have emphasized the main objective of this paper and the role of rSHUD. We've also included a brief description of the SHUD model. Readers can explore the two open demo source codes provided in this paper and the three model cases in the SHUD source code (Cache Creek, Heihe, and Qinghai Lake) to learn and test both rSHUD and SHUD.

We add a paragraph in the section 3.1:

"The SHUD source code, data for three exemplary watersheds, and a straightforward result analysis R script are available on GitHub (https://github.com/shud-system/shud, last accessed in August 2023) and as referenced in (Shu, 2023a). The three showcased examples include Qinghai Lake and the Heihe headwater in China, along with Cache Creek in the United States. Users can download the source code package, compile the model, and initiate the simulation. After the simulation, users can execute the provided R script within the RStudio GUI to retrieve simulation results, facilitating subsequent analysis and visualization."

Thank you for your understanding, and we hope our revisions address your concerns.

**The description of the model is quite unclear. Does the model use 1-D or 2-D shallow water equation for surface runoff? Is there any provision for capillary fringe above the saturated zone in subsurface? Does the model simulate both the Hortonian and Dunnian overland flow? How is the infiltration calculated, using**

**Richards equation? How is potential ET and, subsequently, actual ET calculated? I think it is important to provide this information in the paper since a hydrologist would want to know what does this package offer that the other packages do not? In fact, the most difficult part of running a giant hydrological model such as SHUD is that these are somewhat opaque to a beginner and the conceptual comparisons with other existing models are difficult.**

Thank you for your insightful feedback, highlighting the need for a clearer description of the SHUD model's functionalities and its comparison with other models.

Initially, the focus of this manuscript was to detail the functions and workflow of rSHUD, a toolkit designed for pre- and post-processing tasks related to the SHUD model. The assumption was that potential users would already be familiar with the SHUD model's basics before utilizing the rSHUD package. However, I recognize the importance of providing a concise overview of the SHUD model for the broader audience and the necessity of understanding its unique features.

To address your concerns, I have added a new paragraph in section 2.1, detailing the mathematical and algorithmic structure of the SHUD model, following the outline of your questions. Here's a brief overview:

*"For more detailed explanation and mathematical representation, readers can refer to Shu et al. (2020). We brief the four crucial processes in the watershed hydrology :*

*    - Surface Water Partitioning: While Hortonian and Dunnian overland flows are common assumptions in conceptual models, Integrated Surface-Subsurface Hydrological Models (ISSHMs) like SHUD adopt a more physical description. Instead of these assumptions, they use the Darcy-Richards equation, such as the Green-Ampt method. In SHUD, surface runoff is calculated using Manning's formula for a hydrological computing units (HCU) .*

*- Evapotranspiration: potential evapotranspiration (PET) is computed using the Penman-Monteith equation, while Actual evapotranspiration (AET) is derived by multiplying PET with a soil moisture stress coefficient, determined by soil moisture content and groundwater table depth.*

*- Subsurface flow: Once water infiltrates the ground, it first moves vertically in the Unsaturated Zone. The flux from unsaturated zone to the saturated zone is termed as "recharge to groundwater." The calculation of horizontal groundwater flux in three dimensions is based on the Dupuit Assumption. "*

I genuinely appreciate your feedback, which has undoubtedly enhanced the clarity and comprehensiveness of the manuscript.

**The model estimates soil hydraulic parameters using pedotransfer functions (PTFs)? Are the PTFs used by the rSHUD package universal in some sense? Also, would the PTFs be able to account for macropores which can be very important for percolation?**

Thank you for raising the pertinent question about the universality and capabilities of the pedotransfer functions (PTFs) used by the rSHUD package.

Pedotransfer functions, being empirical equations derived from specific regional laboratory data, inherently possess limitations in their universal applicability. This is evident from the multiple optional PTFs provided by tools like Rosetta software (Zhang et al., 2017; Schaap et al., 2001).

In rSHUD, we have adopted the PTFs from Wosten et al. (2001). However, it's crucial to note that users have the flexibility to select and implement alternative PTFs. Both rSHUD and SHUD are designed to accept parameters from any customized sources.

The primary value of PTFs in our context is to offer an initial estimation of essential hydraulic parameters, such as hydraulic conductivity and

porosity. While uncertainties in these parameter values are acknowledged, the derived parameters still capture the geospatial trends, which significantly influence hydrological processes and the resulting streamflow.

Regarding macropores, their hydraulic characteristics are indeed vital for percolation processes. However, in the absence of a reliable relationship or equations linking soil texture to macropore hydraulic properties, macropore parameters are derived by multiplying a coefficient (default value: 100) with soil matrix properties.

You've rightly pointed out that the combination of soil matrix and macropore properties plays a pivotal role in determining infiltration/percolation processes. It's essential to emphasize that the primary goal of rSHUD is to generate usable input files. It is not designed to provide the "best" or "true" parameters for modeling, a challenge that might be beyond the scope of any pre-processing tool.

Therefore, we revised the paragraph as:
"The original attributes in the soil data include soil texture components such as silt, sand, and clay percentages, as well as bulk density and organic matter content. To derive hydraulic parameters such as hydraulic conductivity, porosity, and van Genuchten parameters, a pedotransfer function is used based on the soil texture data (Wösten et al., 2001; Shu et al., 2020). Pedotransfer functions, being empirical equations derived from specific regional laboratory data, inherently possess limitations in their universal applicability. Users have the flexibility to select and implement alternative PTFs. The primary value of PTFs in rSHUD is to offer an initial estimation of essential hydraulic parameters, while uncertainties in these parameter values should be considered. The pedotransfer function used in rSHUD is listed in Appendix C."

We appreciate your feedback, which has enriched our understanding and presentation of the model's capabilities.

**This brings another crucial point: Most of the effort in hydrological modeling is in calibration of the model parameters. This issue has not even been discussed in the paper.**

Thank you for highlighting the significance of model parameter calibration in hydrological modeling.

I concur with your observation. Calibration is indeed a pivotal aspect of hydrological modeling. However, due to the inherent complexity and importance of calibration, it was deemed beyond the scope of this particular paper. Instead, it warrants a dedicated discussion in a separate article. The intricacies associated with Integrated Surface-Subsurface Hydrological Models (ISSHMs) mirror the complexities faced during the calibration of such models. With over 30 adjustable parameters in the SHUD model, the task of parameter optimization involves navigating a 30-dimensional space to find a globally optimal solution. Given the computational efficiency of ISSHMs, such an extensive search in high-dimensional space is resource-intensive, contributing to the relatively limited application scope of ISSHMs. This high dimensionality also accentuates the equifinality issue.

Furthermore, current distributed hydrological models predominantly employ a global multiplication or addition approach for parameter optimization, which doesn't cater to the spatial heterogeneity of parameter distribution.

In our experience with the SHUD model, the Covariance Matrix Adaptation Evolutionary Strategy has proven to be a reliable and efficient method for calibration. Typically, with 100 trials per generation and over ten generations of parameter optimization iterations, the model can identify an optimized parameter combination. However, the issue of equifinality remains evident.

Further testing, validation, and development efforts are required to address the challenges of parameter optimization in ISSHMs or SHUD specifically.

I appreciate your feedback, which underscores the need for a more in-depth exploration of calibration methodologies in the context of SHUD and similar models.

**Then, there are uncertainties associated with the modeling and hydrological data which are known to the hydrological community for the last 40 years. Should not a package like rSHUD which aims to make data pre-processing easier provide a program for accounting of uncertainties in data?**

I agree the arguments from the reviewer. We revised the descriptions in the manuscript.

We updated the Fig. 8 with latest simulations which is consistent with the data/code uploaded on Zenodo (Shu, 2023b, https://doi.org/10.5281/zenodo.8104324).

Still, the results showed in Fig. 8 is not perfect.  Based on our experiences, there are two reasons: (1) parameters and (2) initial conditions.

In the Fig. 8(b) we can see that the discharge from the watershed is relative small, compared to precipitation and evapotranspiration (ET), leading to the continues water accumulation in storage.  Even though the daily discharge varies largely, the monthly sum of the discharge is very small relative to the precipitation and ET. Therefore, when the curve of monthly discharge is flattened in the water balance plot.

Because of the unknown initial conditions for each triangular units, including the groundwater level, soil moisture and etc., the initial conditions are arbitrary guess (default guess of groundwater storage and soil moisture are 40% of the aquifer). That is the reason that we need a spin-up period to eliminate the error from initial conditions. Or, a modeler can assign the know initial conditions to the model to shorten

the spin-up period. The pronounced recession is due to the error of initial condition.

We do know the problematic results and the error from initial condition are obvious drawbacks. We chose to show the drawbacks rather than to hide them with skillful adjustments. These drawbacks do show the preliminary and raw outputs from the model. Also, such imperfect outputs generally is the outcomes from the first run of hydrological models, the beautiful and acceptable plot generally is the very output of many trial-adjustments or calibration.

Secondly, before the ISSHM, the pre-processing of hydrological models are straightforward and easy-to-handle. But the pre-processing of ISSHM are different story, especially the domain decomposition processes. To realized the optimal computational efficiency and ideal spatial resolution, a modeler using ISSHM have to spent lots of resources and time to repeat the pre-processing and model testing before settle down a set of model input files. That is also one of reason why ISSHM has less applications than other simpler models. Therefore, the team of ISSHM developed tools to implement the model deployment, for example, PIHMgis for PIHM, and parflowio  for ParFlow.

The script for the SHCZO is simple for demonstrate the simplest model deployment, therefore the model's output of SHCZO is very simple and similar to the visualization of Waerma watershed. So we put the simulated results of SHCZO in the appendix E1.

Thank you for your understanding, and we hope our revisions address your concerns.

**The way the paper is written, it is unclear at many places what parts of the modeling process are automated and what parts the user needs to take care of?**

Thanks for your comments.

In response to your feedback, we have made revisions to the manuscript to provide a clearer delineation of the processes facilitated by rSHUD.

To deploy the SHUD model for a specific watershed, a modeler primarily needs to review and adjust the R script tailored for either SHCZO or Waerma. The primary adjustments involve specifying data inputs and setting modeling preferences. Specifically:

Data Inputs: This encompasses the path to the DEM, watershed boundary, river network, soil class, land cover, soil attributes, land cover attributes, and meteorological data.

Modeling Preferences: These settings allow customization of the model's spatial resolution (or the maximum area of a triangle), the maximum number of triangular elements, the minimum angle of a triangular element, boundary simplification tolerance, river simplification tolerance, and the simulation period.

Once these inputs and preferences are set, the script facilitates the model deployment. Users can then run the SHUD model to obtain the desired results.

For result analysis, the script is designed to set up the project environment, indicating the project name and specifying the paths for input and output folders. Subsequent functions within the script enable reading of input and output files, allowing users to further manipulate and analyze the data as needed.

We hope these revisions provide a clearer understanding of the balance between automation and user input within the rSHUD framework. We appreciate your insightful feedback and have made the necessary clarifications in the manuscript.

**Specific comments:**

**Line 47: mathematical equations**

Thank you. I update the sentence.

**Line 49: Also, for hypothesis testing**

Thank you. I update the sentence.

**Line 61-64: Should completely automated modeling even be the aim of process-based modeling given that we want to understand our watershed's functioning which is not possible by ML/DL methods? Note that DL models are still based on correlations and as such do not yield much understanding about physics. Also, given the uncertainties in data which are dominantly epistemic, a fully automated procedure does not even seem like a scientifically sound goal. So, should not the rSHUD package contain some program for data pre-processing so that the user is able to take care of these epistemic uncertainties?**

Yes, the automated procedure is not a scientifically sound goal, as every tiny step matters for scientific research, but it is of practical and engineering value. Our aim with rSHUD is not to advocate for a complete shift towards automation but rather to provide a tool that can streamline certain aspects of the modeling process, especially for scenarios where rapid deployment is essential. For instance:

Ecologists/Agrologists: They might require extensive hydrological modeling outputs across multiple watersheds to support coupled water-ecology or water-agriculture studies.

Decision Makers: In the context of disaster preparedness, there's a need to rapidly deploy regional hydrological models to assess potential flood risks, inundation areas, or other water-driven hazards.

Such needs have been documented in the literature, and many researchers continue to work on tools that facilitate swift model deployment.

However, we acknowledge the importance of addressing epistemic uncertainties. The rSHUD package, while offering automation capabilities, is designed to be modular. Each step in the automated script is composed of individual functions. This design allows modelers to execute specific functions independently, delve into the details, and even modify them as needed since the source code is openly accessible.

In essence, while rSHUD offers automation, it does not limit the researcher's ability to engage in detailed, step-by-step data processing and model building. We believe this balance between efficiency and detailed scrutiny is crucial, and we have made efforts to emphasize this in the revised manuscript.

Thank you for bringing this to our attention, and we appreciate the opportunity to clarify our stance on this matter.

**Line 132: solver**

Thank you. The typo is fixed.

**Line 146: Really? I am not sure what you mean by 'river outlets at the edge of the watershed'?**

When we mention that a river outlet must be located at the edge of the watershed, we are referring to the necessity for the river outlet to be positioned in such a way that it facilitates the drainage of water from the lowest point in the watershed, ensuring there's no artificial accumulation of water. This is the result of data processing.

[Figure]

To better illustrate this:

Subfigure (a): Here, the river outlet is situated within the watershed boundary but does not connect with the lowest element (shown in pink). This configuration leads to water accumulating in the lowest element, creating a pseudo-lake, as water cannot flow into the river or exit the watershed.

Subfigures (b) & (c): In these scenarios, the river outlet is in direct contact with the lowest element. This ensures that water from the lowest element can seamlessly flow into the river, allowing the river to effectively drain water out of the watershed.

The positioning of the river outlet is crucial to prevent unintended water accumulation and to ensure accurate hydrological modeling. We've

included a figure in the manuscript to visually represent these conditions, which should provide a clearer understanding of the concept.

We hope this explanation provides clarity on the matter. We appreciate your feedback.

**Line 161: What slope? Slope of the phreatic surface or the bedrock?**

Thank you for pointing out the ambiguity regarding the term "slope" in Line 161. We acknowledge the oversight and have taken steps to clarify the context.

To address this, we have revised the sentence from Line 159-161 to be more explicit:

"Accordingly, the river network of the SHUD model is superimposed upon triangular cells. This configuration facilitates the computation of groundwater exchange between hill-slopes and the river network. The exchange is determined by the hydraulic gradients existing between the river channel and the groundwater level. Meanwhile, surface fluxes are calculated utilizing the weir flow equation."

**Line 171: Combination of functions**

Thank you. The grammar issue is fixed in revised manuscript.

**Line 185: What is mean by 'reference data'?**

Thank you.

By "reference data," we are referring to datasets that are used for calibration and validation purposes. These datasets are considered as

realistic or acceptable estimations of the variables under study. The most common type of reference data in hydrological modeling is observational streamflow. However, there are other datasets that can be used as reference data, especially in the context of ungauged watersheds. Examples include evapotranspiration derived from remote sensing, groundwater levels from monitoring wells, and river stages obtained through remote sensing combined with deep learning techniques. While these datasets might not be as accurate as in-situ streamflow data, they provide valuable insights and are particularly useful for calibration in regions where traditional data might be scarce or unavailable.

We have incorporated this explanation into the manuscript to provide clarity on the term and its significance in the context of the study.

**Line 207: Is user supposed to standardize these files? Any guidance on how to do it?**

Thank you.

In the rSHUD package, we utilize the xts data class in R, which is specifically designed for handling time-series data. This class supports a matrix format where each row corresponds to a specific time point and columns represent different variables. For instance, in the context of meteorological forcing data, columns could represent variables such as precipitation, temperature, wind speed, radiation, and air pressure.

While users might have their data in various original formats, our package provides the write.tsd() function to facilitate the conversion of these datasets into the specific format required by the SHUD model. The output generated by the write.tsd() function is designed to be self-explanatory, allowing users to easily understand the structure. Moreover, if users wish to save the standardized data in other formats, they can utilize other software or programs, leveraging the clear structure provided by our output.

We have expanded upon this in the manuscript to offer users a clearer understanding of how to standardize their time-series data for use with the SHUD model.

**Line 215: "It is necessary that ... can occur otherwise".**

Thank you for your suggestion. We rewrite the sentence:

"It is necessary to ensure the accurate overlay of data from multiple sources to prevent potential spatial inconsistencies."

**Line 241: What about the depth to bedrock?**

Thank you.

The depth to bedrock (or aquifer thickness) is not involved in the triangulation process. The depth of bedrock is used in function shud.mesh(), that build the prism of hill-slope element.

**Line 251: 'the' instead of 'The'**

Thank you. The typo is fixed in the revised version.

**Line 275: Any references for this?**

Thank you for your reminder. We add the necessary citations in the revised version.

The river order implies the contribution area and geometry. Actually, the strong corelationship exist among slope, geometry (width, depth, area of the cross-section), length, contribution area, river order, discharge and so on (Flint, 1974; Kratzer et al., 2006; Downing et al., 2012; Perron and Royden, 2013; Strick et al., 2018; McManamay and DeRolph, 2019).

Flint, J. J. (1974). Stream gradient as a function of order, magnitude, and discharge. Water Resources Research, 10(5), 969–973. https://doi.org/10.1029/WR010i005p00969

Kratzer, J. F., Hayes, D. B., & Thompson, B. E. (2006). Methods for interpolating stream width, depth, and current velocity. Ecological Modelling (Vol. 196). https://doi.org/10.1016/j.ecolmodel.2006.02.004

Downing, J. A., Cole, J. J., Duarte, C. M., Middelburg, J. J., Melack, J. M., Prairie, Y. T., et al. (2012). Global abundance and size distribution of streams and rivers. Inland Waters, 2(4), 229–236. https://doi.org/10.5268/IW-2.4.502

Perron, J. T., & Royden, L. (2013). An integral approach to bedrock river profile analysis. Earth Surface Processes and Landforms, 38(6), 570–576. https://doi.org/10.1002/esp.3302

Strick, R. J. P., Ashworth, P. J., Awcock, G., & Lewin, J. (2018). Morphology and spacing of river meander scrolls. Geomorphology, 310, 57–68. https://doi.org/10.1016/j.geomorph.2018.03.005

McManamay, R. A., & DeRolph, C. R. (2019). A stream classification system for the conterminous United States. Scientific Data, 6, 190017. https://doi.org/10.1038/sdata.2019.17

**Line 285: Explain how the hydraulic parameters are generated.**

Thank you for your suggestion.

The hydraulic parameters for river channel include the sinuosity coefficient of the river reach, manning's roughness, weir flow coefficient, conductivity and thickness of sediment. These parameters control the open channel flow, surface and subsurface exchanges between hill-slope and river reach. Since the correct values of these parameters are unknown during the model deployment, the shud.river()

program generate the default values as initial guess. Users still can modify the value based on their measures.

The sentences were rewritten as:

"The hydraulic parameters of the river channel encompass five components: sinuosity coefficient of the river reach, Manning's roughness, weir flow coefficient, conductivity and thickness of sediment. These parameters are pivotal in dictating open channel flow and the exchange dynamics between the hill-slope and the river reach. Given the inherent uncertainties and the lack of precise values during the initial model deployment, the \emph{RiverType} function provides default values as an initial guess. However, it is worth noting that users retain the flexibility to modify these values based on their measurements or other reliable sources."

**Line 288: What do you mean by 'overlapping relationship'?**

Thanks for your reminder.

The topological relationship between a river reach and a triangle is defined as intersect. We updated the terminology in revised manuscript.

**Line 300: a cell**

Thank you. The typo is fixed in the revised version.

**Line 308: 'Vertical flux of water from the surface to the soil' is infiltration? Why not just write infiltration?**

**Also, adding a table of all the model parameters would be really helpful for the readers.**

Thank you. We use the "infiltration" in the revised version.

We added a list of all parameters in the appendix and a description of the parameters in the section 4.2.4.

**Line 317-324: Clarify what the user needs to do to tackle this issue?**

Thank you.

Land cover classifications can vary depending on the dataset being used. To address this, we've provided default look-up tables for three widely-used global/regional land cover classification systems in Appendix A. These tables are based on established literature and datasets. However, we recognize that users might have specific needs or might be using different classification systems.

To tackle this, users can modify the values in the provided look-up tables as needed. If they are employing a different classification system not covered by our default tables, they would need to develop a new look-up table tailored to that system. We recommend using the three provided land cover look-up tables as a reference to ensure consistency and accuracy in the classification process.

**Line 327: reads**

Thank you. The typo is fixed in the revised version.

**Line 332: What is 'experiential'? So, the melting factor stays the same for each day in a month? Would this not introduce any artifacts at the end of the month where the melting factor suddenly changes?**

Thank you for pointing out the typo and raising a pertinent question regarding the snow melting factor.

Firstly, the term "experiential" was indeed a typographical error and should have been "empirical." We apologize for the oversight and have corrected it in the revised manuscript.

Regarding your observation about the melting factor (MF) remaining constant for each day within a month: you're correct. In our current implementation, the MF remains consistent throughout the month and then undergoes a sudden change at the start of the subsequent month. This approach is based on empirical methods where, in some models, the MF is even held constant throughout the entire year. It's a limitation inherent to the empirical degree-day method of computing snow melting.

**Equation 1: Subscripts in the equation should not be italic as these are not variables. Same comments apply for equation in the Appendix B.**

Thank you. We have taken your feedback into account and have retyped the equation in the revised manuscript to ensure that the subscripts are not italicized, adhering to the standard notation conventions.

**Line 341: But a modeler would always want to and should use the observational data to constrain the model if such data are available? Your results in Figure 8 show that the parameters used are not adequate. Why are the results for SCHZO not shown?**

Thank you.

You're right; when available, observational data is invaluable for constraining and calibrating models. The results in Figure 8 were presented to demonstrate the model's functionality and the workflow of rSHUD. We acknowledge that without calibration, the simulated discharge might not align perfectly with observations.

Regarding the SHCZO results, they were not included to maintain brevity, as they were similar to the Waerma results. However, for the sake of completeness, we can consider adding them to an appendix in the next revision if deemed necessary.

**Line 406: What do you mean by 'expected minimum resolution of the modeling'?**

Thank you for seeking clarification on the term 'expected minimum resolution of the modeling' mentioned in line 406.

In the process of constructing the Waerma watershed model, our ideal maximum triangle area was set to $150\ m \times 150\ m = 22500\ m2$ . The equivalent resolution is 150m, which is derived by taking the square root of the unit area. With a maximum unit area of 22500 m^2, the coarsest (or minimum) resolution is 150m. Therefore, in our R script for constructing the Waerma model (waerma.sh), we configured `a.max = 150*150`.

I hope this provides clarity on the concept. We'll consider revising the manuscript to elucidate this point further.

**Line 409: Thiessen polygons are generated for interpolating meteorological data, right? If so, mention this. Also, I am assuming that the meteorological data are used in distributed manner by the model, right?**

Thank you for your observation regarding the use of Thiessen polygons.

To clarify, the Thiessen polygons in this context are not utilized for spatial interpolation of meteorological data. Instead, they are employed to delineate the coverage area for each meteorological station. In both rSHUD and SHUD, we do not perform spatial interpolation on

meteorological data but directly use their time series. For instance, if we assume there are N triangles falling within the coverage of the 1st Thiessen polygon, then the .sp.att file (fig. 5a) will assign a Forcing_ID of 1 to these N triangles. This indicates that the meteorological forcing data for these N triangles are provided by TSD of the 1st Thiessen polygon.

We will ensure to make this distinction clear in the manuscript to avoid any confusion.

**Line 439: Not clear what you mean by other spatial analysis and hydrological data processing?**

Thank you for seeking clarity on the phrase "other spatial analysis and hydrological data processing" mentioned in line 439.

The rSHUD package is comprehensive, comprising 163 functions. While we've highlighted only a select few in the manuscript, these functions cater to a wide range of data analysis and processing needs beyond what's explicitly mentioned. For example:

The fishnet() function is designed to generate a regular grid, which can be in the form of polygons, polylines, or points.

The MeshData2Raster() function facilitates spatial interpolation, producing a raster file based on values derived from triangular cells.

The FDC function is tailored to generate the flow duration curve of discharge.

For a comprehensive understanding of all the functions in the rSHUD package, users can refer to the rSHUD help files. As mandated by R, each function comes with a detailed description. By typing help(FunctionName), users can access these descriptions. Additionally,

the command ls("package:rSHUD") provides a list of all functions available within the rSHUD package.

We appreciate your feedback and will ensure to elaborate on this aspect in the manuscript for clearer understanding.

**Line 445: archived**

Thank you. The typo is fixed in the revised version.

---

## Referee Report (RR1)

[referee-annotated manuscript omitted]

---

## Author Response (AR2)

**To Reviewer Comment:**

Following are my reply to the comments.

The reviewer's comments and questions are in bold, and my reply is in blue and normal text.

**Line 49:**
Typo is fixed. Thank you.

**Line 50:**
Typo is fixed. Thank you.

**Line 59: amounts of modeling??**
The sentence is written as: "GUI interface tools face difficulties in handling large amounts of modeling tasks."

**Line 60-61:Rewrite this sentence. Also, What do you really want to say here? Human participation is indispensable using GUI-based modeling or is it indispensable in general.**
Human intervention decreases reproducibility, as different modelers may handle the data in various ways. For instance, the watershed delineation by different modelers can yield different results. Conversely, the results from automation always depend on parameters and rules, which are reproducible in different environments.

The sentence is rewritten as: "In modeling with GUI-tools, human participation is required instead of being controlled via modeling parameters, making it impossible to implement large number of hydrological modeling cases. "

**Line 66: But in the previous paragraph you say that it is impossible to do so? Please be clear.**

I meant the GUI-tools cannot do such jobs, so we need intervention-free, reproducible, and automated tools.

The sentence is rewritten as: "Therefore, when dealing with a large number of watershed simulation tasks, both pre-processing and simulation post-processing necessitate the utilization of intervention-free, reproducible, and automated tools."

**Line 67-68: Write something about epistemic uncertainties here, and that a modeler needs to take care of these uncertainties according to their knowledge of the data. You can refer to several recent papers by Keith Beven to make these arguments. For example, see Beven (2019) in Porceedings of Royal Society**

Thank you. We add following sentences:

"However, regardless of the modeling tools used, it encompasses epistemic uncertainty, affecting the final results and accuracy of hydrological simulations(Beven and Young, 2013; Beven, 2013, 2018, 2019). Various modeling tools can technically support the modeling process but cannot eliminate the inherent uncertainties in models and data. Users need to be vigilant about this"

Beven, K., 2013. So how much of your error is epistemic? Lessons from Japan and Italy. Hydrol. Process. 27, 1677–1680. https://doi.org/10.1002/hyp.9648

Beven, K.J., 2018. On hypothesis testing in hydrology: Why falsification of models is still a really good idea. WIREs Water 5, 1–8. https://doi.org/10.1002/wat2.1278

Beven, K., Young, P., 2013. A guide to good practice in modeling semantics for authors and referees. Water Resour. Res. 49, 5092–5098. https://doi.org/10.1002/wrcr.20393

Beven, K., 2019. Towards a methodology for testing models as hypotheses in the inexact sciences. Proc. R. Soc. A Math. Phys. Eng. Sci. 475. https://doi.org/10.1098/rspa.2018.0862

**Line 93:**
Typo is fixed. Thank you.

**Line 147: What is unphysical about Hortonian and Dunnian flows? They both occur nature.**

Here, we need to define what "physically" means. Physically means (1) laws are based on minimal assumptions and (2) these laws are applicable in both laboratory and field settings (at least in most scenarios). Both Hortonian and Dunnian are conceptual models for ideal slope runoff generation, and they cannot effectively apply under various slope gradients, soil textures, vegetation, and precedent soil moisture conditions. Detailed discussions on these two methods and similar mechanisms can be found in Beven (2012) (section 1.4, page 4-13, figure 1.4). In fact, both Hortonian and Dunnian mechanisms are conceptual models or expedients for calculating overland flow, rather than universally applicable physical laws. In contrast, the Darcy-Richards equation used in soil water calculations and the St Venant equation used for surface flow are based on fewer assumptions and have better adaptability on different slopes, hence can be considered as "physical." This also explains why the Darcy equation is sometimes referred to as Darcy's Laws. However, it should be noted that outside the scope of watershed Rainfall-Runoff models, the Darcy equation may not be an absolute truth.

**Line 148: GA method is actually based on approximate theory which is surely not how water moves in soils.**
The GA method is not a perfect approach for calculating infiltration in the topsoil. Although it is one of the physical methods that can be employed in a temporal-spatial continuum like numerical schemes. The foundation of the GA method still relies on Darcy's equation, assuming a sharp wetting front and uniform soil conductivity. It has its limitations. However, there is currently no purely physical equation or law that adequately describes the continuous vertical flux from the topsoil to the unsaturated zone and eventually to the saturated zone. Such a law should encompass the temporal-spatial continuum and account for preferential/macropore flows. The GA method is not a purely physical or realistic representation, but it serves as a more physical and practical option for numerical methods.

**Line 149:**
Typo is fixed. Thank you.

**Line 149:May be I am missing something here. But surface runoff is calculated using rainfall and infiltration runoff; the Manning's equation is used to compute flow height from the runoff .**
In the SHUD model, the infiltration is calculated based on the height of ponding water on the land surface.  The lateral runoff is calculated based on the residual water on surface after infiltration, in which the Manning's equation is applied.

**Line 150:**
Typo is fixed. Thank you.

**Line 154-155: Again, If I am not wrong, Dupuit assumptions are used to approximate three dimensional flow as two or one dimensional flow.**

Thank you.

The sentence is rewritten as: "The calculation of horizontal groundwater flux in horizontal direction is based on the Dupuit Assumption"

**Line 278:**

Typo is fixed.

**Line 279: "The substantial difference in triangle area between the edges and inner parts significantly slows down model performance by unnecessarily increasing computations on the boundary"  ?? how??**

For example, the Waerma watershed has a total area 9.8 km2, whose boundary was built on the 30-meter DEM. Then the minimum line segments on the boundary is 30-meter.  When we try to built a triangular mesh with ideal area 0.1 km2, the expected area of each triangle is 0.1km2 approximately.  The actual number of triangles may be more than 98 (9.8 / 0.1).

Here is two option:

1) To generate the mesh directly with the boundary from 30-meter DEM.

2) To simplify the boundary with tolerance 60-meter, then generate the mesh.

The results are shown in following figure:

**(a) No Simplification                    (b)Simplified (60m)**

[Figure]

[Figure]

There are lots of small triangles near the boundary in the non-simplified boundary. These emergence of such triangles is due to the 30-meter boundary jigsaw. The the number of triangles in two scenarios are 1463 and 160 respectively.

The following R code can repeat the experiment.

```
clib=c('rgdal', 'rgeos', 'raster', 'sp', 'fields', 'xts')
x=lapply(clib, library, character.only=T)
library(rSHUD)

data("waerma")
wbd=waerma$wbd
x=spTransform(wbd, CRSobj = crs.Albers(wbd))
x1=x
x2=rgeos::gSimplify(x, tol=60)
amax = 1e5 #km2
y1=shud.triangle(wb=x1, q=30, a=amax)
y2=shud.triangle(wb=x2, q=30, a=amax)
png('vs.png', width = 7, height=4, res=400, unit = 'in')
par(mfrow=c(1, 2))
plot(y1, type='n', asp=1, main='(a) No Simplification')
plot(y2, type='n', asp=1, main='(b)Simplified (30m)')
dev.off()
print(nrow(y1$T))
print(nrow(y2$T))
```

**Line 299: Higher of the orders of the two joining streams is used.**

Thank you. The sentence is rewritten as: "In cases where two streams of different order join, the higher of the orders of the two joining streams is used."

**Line 309-310:Unclear. Are you not calculating the average of slopes of small segments within the reach?**

The calculation of river reach slope is the elevation differences and distance between the start and end points, instead of the average of slope of each segments within the reach.

**Line 333: What do you mean by 'may contain'? Typically, a cell is a homogeneous unit right?**

Thank you. The sentence is rewritten as: "a cell contains only one soil type without heterogeneity within the cell."

**Line 334-335: So, a cell is a homogeneous unit.**

Yes. A cell is a homogeneous unit.

**Line 364: TSD?**

TSD stands for Time-Series Data. The first TSD is in Line 196.

**Line 370: Should it not vary depending upon the if the year is leap year or not?**

Thank you. At the end of the paragraph, we add a sentence:

"The denominator 366, which represents a leap year, is replaced with 365 for a common year."

**Line 476: There should be a brief discussion of the data uncertainties because estimating uncertainties in data is a pre-processing task. It would be prudent to make reader aware that the need to consider uncertainties in data on their own and it is not done by the rSHUD package.**
Thank you.

We add a paragraph:

"Uncertainty is crucial in hydrological modeling and must be considered even when using the rSHUD package for model deployment. Users should acknowledge uncertainties in data inputs and model parameters, which may arise from measurement errors, natural variability, or limitations in the package's model structure and parameterization. The equations and data embedded within rSHUD package also introduce uncertainties. Users of the rSHUD package should therefore conduct a thorough uncertainty analysis as a preprocessing step to ensure the reliability and robustness of their modeling outcomes."

**To Polina Shvedko Comment:**

**It seems that table is included as figure #5. If it is so, it must be re-labelled as table and the references in the manuscript text must be adjusted accordingly.**

We confirmed. We labeled the figure 5 on purpose. Even though the content in figure 5 are tables, but they represent the linkage of the files, so the figures are preferred.